# Magnet: We Never Know How Text-to-Image Diffusion Models Work, Until We Learn How Vision-Language Models Function

**Chenyi Zhuang**[1]    **Ying Hu**[1]    **Pan Gao**[1,2*]

[1] Nanjing University of Aeronautics and Astronautics
[2] Key Laboratory of Brain-Machine Intelligence Technology, Ministry of Education
`{chenyi.zhuang,ying.hu,pan.gao}@nuaa.edu.cn`

## Abstract

Text-to-image diffusion models particularly Stable Diffusion, have revolutionized the field of computer vision. However, the synthesis quality often deteriorates when asked to generate images that faithfully represent complex prompts involving multiple attributes and objects. While previous studies suggest that blended text embeddings lead to improper attribute binding, few have explored this in depth. In this work, we critically examine the limitations of the CLIP text encoder in understanding attributes and investigate how this affects diffusion models. We discern a phenomenon of attribute bias in the text space and highlight a contextual issue in padding embeddings that entangle different concepts. We propose **Magnet**, a novel training-free approach to tackle the attribute binding problem. We introduce positive and negative binding vectors to enhance disentanglement, further with a neighbor strategy to increase accuracy. Extensive experiments show that Magnet significantly improves synthesis quality and binding accuracy with negligible computational cost, enabling the generation of unconventional and unnatural concepts. Code is available at `https://github.com/I2-Multimedia-Lab/Magnet`.

## 1 Introduction

Recently, Text-to-Image (T2I) diffusion models [1, 2, 3, 4] have drawn considerable attention from both the research community and industry. Among these models, Stable Diffusion (SD) [2] uses the CLIP text encoder [5] to encode the given prompt, which is relatively lightweight than other diffusion models that adopt T5 [6]. Unfortunately, generating text-aligned images is still challenging for SD and requires multiple runs to achieve the desirable results. Several works [7, 8, 9, 10] have pointed out that the blended context by the CLIP text encoder causes improper binding. However, few have analyzed in detail how the text encoder affects the generation of the diffusion model.

Step back and refocus on the CLIP text encoder—an integral part of the Vision-Language Model (VLM). Prior studies [11, 12] have observed VLMs lacking compositional understanding and investigated them on image-to-text retrieval benchmarks. In this work, we are motivated to answer **how the text encoder understands attribute**, and **how it affects the attribute binding of T2I diffusion models**. Upon closer inspection, we observe a phenomenon of attribute bias and discern a contextual problem in padding embeddings, leading to a well-known T2I issue—concept bleeding [10].

Based on the observation, we introduce the binding vector, which is applied to the text embedding of each object. With positive and negative binding vectors, each object can pull target attributes and push unrelated attributes to distinguish them from each other. We further introduce a neighbor strategy to ensure an accurate estimation of the binding vector. Our manipulation is performed strictly in the textual space, without training, fine-tuning, or additional datasets and inputs. Overall, the main

---

*Corresponding author

38th Conference on Neural Information Processing Systems (NeurIPS 2024).

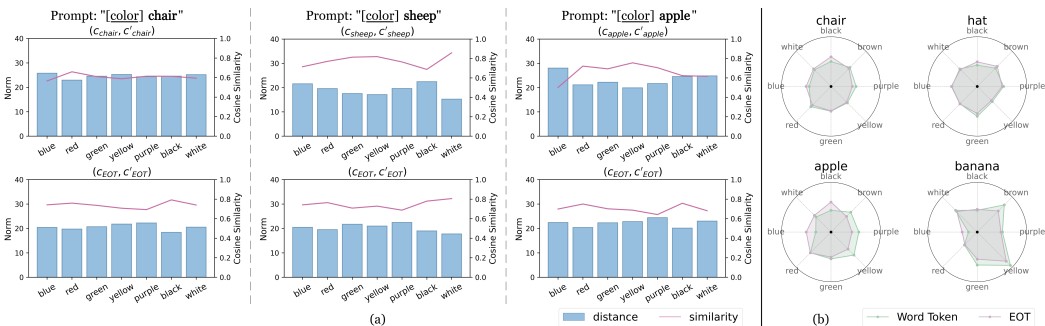

Figure 1: Analysis of the CLIP text encoder for understanding attributes. There is a discrepancy between the word and [EOT] embeddings of the attribute bias on different objects.

contributions of this work are: (1) We highlight a contextual issue in the text encoder, which impacts diffusion-based image generation. (2) We propose a novel training-free method to address the binding problem. (3) Extensive experiments are conducted to verify the effectiveness of Magnet.

## 2    Analysis of the CLIP text encoder and the diffusion model

In this section, we aim to recognize the pattern of the CLIP text encoder for understanding attributes, then go deep into the diffusion model to analyze the underlying reason for improper attribute binding.

The CLIP text encoder uses the causal mask mechanism to produce a unidirectional context, i.e., each word can only consider the words to their left [13, 14]. It performs contrastive learning on a specific *End of Text* ([EOT]) embedding without word-level supervision, while prior studies [15, 16, 17] suggest each *word* has a semantic effect on the generated image. In this case, we categorize two types of embedding as **word** and **[EOT]** for fine-grained analysis. Consider a prompt encoded to text embeddings $c = \{c_{SOT}, c_{p_1}, ..., c_{p_N}, c_{EOT}\}$ consisting $N$ word embeddings. Different from the [EOT] embedding $c_{EOT}$, word embeddings $c_{p_1}, ..., c_{p_N}$ are unsupervised during training. We skip the embedding of *Start of Text* ([SOT]) $c_{SOT}$ for simplicity. To study how the two types of embedding understand attributes, we select 60 familiar objects and 7 common colors to obtain text embeddings $c = \{c_{object}, c_{EOT}\}$ and $c' = \{c'_{color}, c'_{object}, c'_{EOT}\}$, i.e., without and with the color context, respectively. We aim to compare: (1) contextualized word embedding $c'_{object}$ with the original $c_{object}$ ; (2) contextualized [EOT] embedding $c'_{EOT}$ with the original $c_{EOT}$.

**How do two types of embedding understand attributes?** Fig. 1 (a) compares the Euclidean distance and the cosine similarity between embeddings with and without the color context. The pattern varies between cases or objects. As to the word embedding, the similarity curve of *"chair"* is relatively smooth, but that of *"sheep"* has a large gap between *"black"* and *"white"*. Similarly, *"blue apple"* diverges from others. The above phenomenon, which we call **attribute bias**, describes the tendency of an object to *favor* certain attributes over others. We compare the attribute bias per object for two embedding types in Fig. 1 (b). If the object has a natural composition in human knowledge, it presents a serious attribute bias (e.g., *"yellow banana"* v.s. *"blue banana"*). Meanwhile, the word embedding is more *volatile* than the [EOT] embedding, which shows *less dramatic* change. Our hypothesis is the absence of word-level supervision during CLIP training, as well as the bags-of-words behavior of VLMs [11, 12]: **the [EOT] embedding is trying to *remember* all important words in the given prompt**, including adjectives and nouns. However, it leads to an inaccurate textual representation of the [EOT] embedding, affecting the interaction between the image latent and semantic word embeddings. We have provided more analysis and examples in Appendix A.1 (see Fig. 12, 13).

**How do two types of embedding affect SD?** In practice, SD pads the input prompt to a fixed length $L = 77$ using additional [EOT] embeddings (i.e., the padding token is initialized by the symbol [EOT]), denoted as $c_{pad_l}$, where $l = 1, ..., L-N-2$. To study the attribute binding during generation, we modify the above two text embeddings $c$ and $c'$ with $L - N - 2$ padding token embeddings, then design 4 fine-grained cases: (1) standard generation conditioned on $c'$ where all text embeddings contain the color context; (2) only replace $c'_{object}$ with $c_{object}$ to eliminate the context on the word embedding; (3) only replace $c'_{EOT}$ with $c_{EOT}$, as well as padding embeddings $c'_{pad_l}$ with $c_{pad_l}$; (4) eliminate the color context on word, [EOT], and padding embeddings. Fig. 2 (a) presents 3 examples,

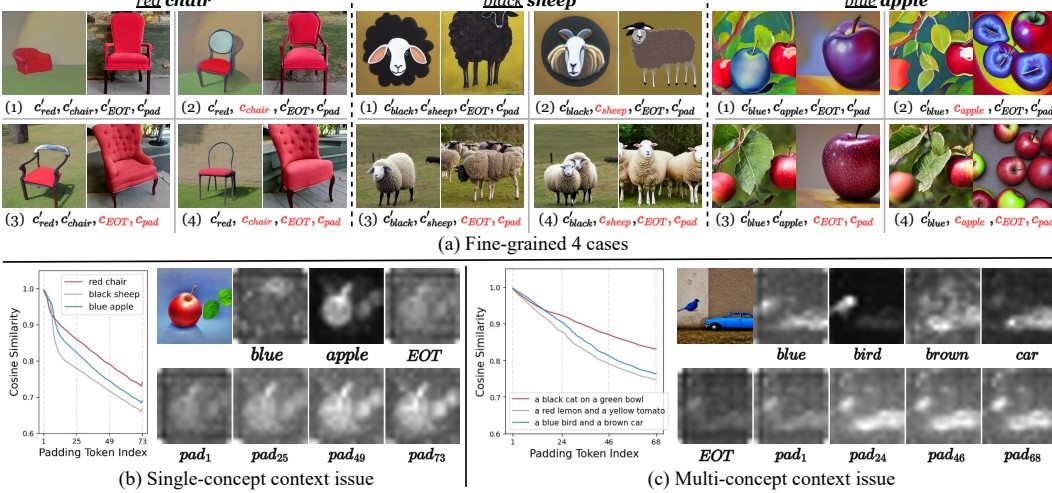

Figure 2: (a) Fine-grained study through our designed embedding swapping experiment. The context issue in padding embeddings for (b) single-concept scenario, and (c) multi-concept scenario.

including a natural concept *"red chair"*, unnatural concepts *"black sheep"* and *"blue apple"*. Cases 1-2 can be observed in all examples with less realistic and painting-like images. **Note that the used SD is trained to generate photo-realistic images**. These results indicate a deviation from the learned distribution. Conversely, cases 3-4 produce realistic images, but neglect the target color when the concept is unnatural. This suggests that the context in [EOT] and padding embeddings do have a significant impact on attribute generation. In Appendix A.2, we describe the above 4 cases in detail and provide more examples, as well as 3 additional cases (see Fig. 14).

**Why improper binding?** We posit that the first [EOT] has a close-knit context with word embeddings. The padding embedding, however, may deviate due to the causal attention mechanism. We then compute the cosine similarity between [EOT] and each padding embedding $l = 1, ..., L$, and dive into their cross-attention activations on single- and multi-concept scenarios. Fig. 2 (b) studies a prompt with only one object. The curve drops more drastically on the unnatural concept *"blue apple"* than the natural one *"red chair"*. Cross-attention shows that *"apple"* overlaps with padding embeddings (e.g., $pad_{73}$) rather than the [EOT] embedding. **It is like these padding embeddings have *forgotten* part of the context remembered in the first [EOT]**. Without interference from other concepts, the inaccurate context in these padding embeddings causes out-of-distribution, or the binding of another attribute if the model learns an underlying bias in the training dataset (e.g., *"apple"* prefers *"red"*). Fig. 2 (c) further studies the context issue on multiple concepts. For two objects *"bird"* and *"car"*, even though the activations of their word embeddings do not overlap, cross-attention shows obvious entanglement in these padding embeddings. This multi-concept context issue in padding embeddings, i.e., entangled concept representations, explains color leakage and object sticking. We refer the reader to Appendix A.3 for a comprehensive analysis of this context issue.

**How to disentangle different concepts?** Prior studies [8, 18] prove that these padding embeddings are essential for image quality and can not simply be removed. Also, it is impossible to manipulate one single concept in these padding embeddings due to their entangled property. On the other hand, these naturally separated word embeddings show editability. For instance, Fig. 2 (a) *"black sheep"* from case 2 to case 1 changes only the word embedding of *"sheep"* while encouraging the desired attribute. We are inspired to manipulate the word embedding of each object, therefore strengthening the binding within each concept and enhancing the distinction between concepts.

## 3 Magnet: disentangling concepts with the binding vector

Our approach is based on two key observations: the context issue of the padding embedding, and the controllability of the word embedding. We introduce the **binding vector**, which can be applied on the object embedding to attract the target attribute and repulse other attributes, analogous to a **Magnet**.

**Preliminary:** Given a prompt $\mathcal{P}$, we use Stanza's dependency parsing module [19] to extract each *concept*, denoted $A\&E$, where $E$ is the object word and its target attribute as $A$. The dependency

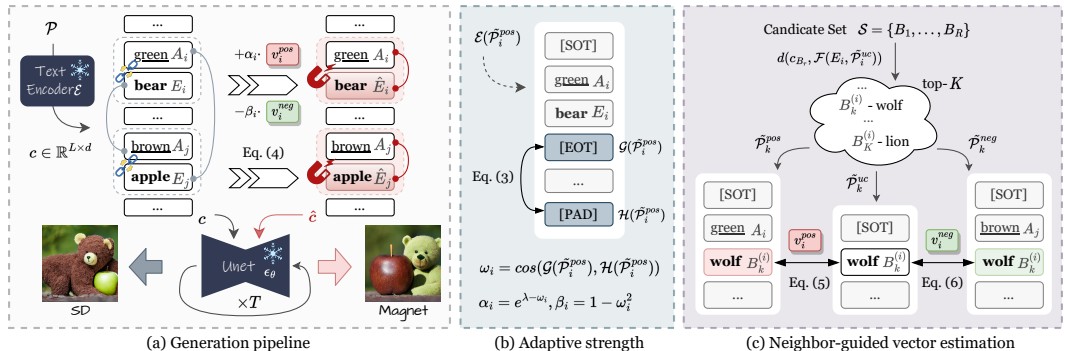

|(a) Generation pipeline | (b) Adaptive strength | (c) Neighbor-guided vector estimation|

Figure 3: Overview of the proposed Magnet. We manipulate the object embedding with the positive and negative binding vectors, which are estimated with the guidance of neighbor objects.

set with $M$ concepts is $\mathcal{D} = \{A_1 \& E_1, ..., A_M \& E_M\}$. Detailed dependency extraction is given in Appendix B.1. Then the pre-trained CLIP text encoder $\mathcal{E}$ is applied to map $\mathcal{P}$ to the text embedding $c = \{c_{SOT}, c_{A_1}, c_{E_1}, ..., c_{A_M}, c_{E_M}, c_{EOT}, c_{pad_1}, ..., c_{pad_{L-N-2}}\}$. For simplicity, we omit the linking words. We treat the diffusion model as a black box and leave its background in Appendix B.2.

For each object $E_i$ in $\mathcal{D}$ with the word embedding $c_{E_i}$, we aim to estimate its positive binding vector $v_i^{pos}$ to pull the target attribute $A_i$, and its negative binding vector $v_i^{neg}$ to push other attributes.

## 3.1 Apply the binding vector on the object embedding

Instinctively, the binding vector can be estimated by the object itself. To be specific, we compose new concepts *out of the current context* of $\mathcal{P}$, which are: (1) unconditional concepts, $\tilde{\mathcal{P}}_i^{uc} = \{\varnothing \& E_i\}$, where $\varnothing$ is a blank text ""; (2) positive concepts, $\tilde{\mathcal{P}}_i^{pos} = \{A_i \& E_i\}$; (3) negative concepts, $\tilde{\mathcal{P}}_i^{neg} = \{A_j \& E_i | j = 1, ..., M, j \neq i\}$. The positive and negative binding vectors are estimated by:

$$v_i^{pos} = \mathcal{F}(E_i, \tilde{\mathcal{P}}_i^{pos}) - \mathcal{F}(E_i, \tilde{\mathcal{P}}_i^{uc}) \tag{1}$$

$$v_i^{neg} = \mathcal{F}(E_i, \tilde{\mathcal{P}}_i^{neg}) - \mathcal{F}(E_i, \tilde{\mathcal{P}}_i^{uc}) \tag{2}$$

where $\mathcal{F}(\cdot)$ extracts the word embedding of the object $E_i$ in a specific *decontextualized* prompt. Note each object has $M - 1$ negative concepts, resulting in $M - 1$ negative binding vectors to punish all unrelated attributes $A_j, j = 1, ..., M, j \neq i$. Note that these positive and negative attributes are prompt-dependent [2]. We introduce the unconditional concept as a pivot to avoid the need for manual definition or semantic contrast between positive and negative attributes.

Based on our analysis of the context issue in the padding embedding in Section 2, we hypothesize an association between the attribute bias and the strength. Intuitively, unnatural concepts (e.g., *"blue banana"*) suffer more attribute bias and their padding embeddings are more tend to *forget* the concept. In this case, we need to manipulate the word embedding significantly to ensure strong binding. We introduce the **adaptive strength** of the binding vector for each object $E_i$:

$$\alpha_i = e^{\lambda - \omega_i}, \beta_i = 1 - \omega_i^2, \quad \text{where} \quad \omega_i = cos(\mathcal{G}(\tilde{\mathcal{P}}_i^{pos}), \mathcal{H}(\tilde{\mathcal{P}}_i^{pos})) \tag{3}$$

where $\mathcal{G}(\cdot), \mathcal{H}(\cdot)$ extract the first [EOT] embedding and the last padding embedding in text embeddings $\mathcal{E}(\tilde{\mathcal{P}}_i^{pos})$, respectively. $\lambda$ is a positive constant. Please refer to Appendix B.3 for the inspiration of the formula, and the statistical analysis for the choice of the hyperparameter $\lambda$.

Finally, the object embedding $c_{E_i}$ in the initial text embeddings $c = \mathcal{E}(\mathcal{P})$ is modified by:

$$\hat{c}_{E_i} = c_{E_i} + \alpha_i \cdot v_i^{pos} - \beta_i \cdot v_i^{neg} \tag{4}$$

## 3.2 Neighbor-guided vector estimation

In practice, we find that using a single object to estimate the binding vector can be inaccurate and fail to disentangle concepts (see Fig. 7 and Fig. 19). In this case, we introduce the **neighbor strategy**

---

[2] The same object in different prompts may have different positive and negative attributes.

to ensure an accurate estimation. These neighbor objects should have similar representations to the target object in the learned textual space. We define a candidate set $\mathcal{S} = \{B_1, ..., B_R\}$ with $R$ objects that has pre-processed to $\{c_{B_1}, ..., c_{B_R}\}$, which is the collection of the word embedding $c_{B_r}$ in $\mathcal{E}(B_r) = \{c_{SOT}, c_{B_r}, c_{EOT}, ...\}, r = 1, ..., R$. The top-$K$ neighbor objects for the target object $E_i$ are determined by $d(c_{B_r}, \mathcal{F}(E_i, \tilde{\mathcal{P}}_i^{uc}))$, where $B_r \in \mathcal{S}$, $d(\cdot)$ denotes the cosine similarity.

In Appendix B.4, we describe this neighbor strategy in detail, and further discuss a way to predict semantic neighbors using pre-trained large language models.

With the selected neighbors of the target object $E_i$, denoted $\{B_k^{(i)}\}_{k=1}^K$, we compose the unconditional concepts $\tilde{\mathcal{P}}_k^{uc} = \{\varnothing \& B_k^{(i)}\}$, positive concepts $\tilde{\mathcal{P}}_k^{pos} = \{A_i \& B_k^{(i)}\}$, and negative concepts $\tilde{\mathcal{P}}_k^{neg} = \{A_j \& B_k^{(i)} | j = 1, ..., M, j \neq i\}$. The estimation of the binding vector is then rewritten as:

$$v_i^{pos} = \frac{1}{K} \sum_{k=1}^K (\mathcal{F}(B_k^{(i)}, \tilde{\mathcal{P}}_k^{pos}) - \mathcal{F}(B_k^{(i)}, \tilde{\mathcal{P}}_k^{uc})) \tag{5}$$

$$v_i^{neg} = \frac{1}{K} \sum_{k=1}^K (\mathcal{F}(B_k^{(i)}, \tilde{\mathcal{P}}_k^{neg}) - \mathcal{F}(B_k^{(i)}, \tilde{\mathcal{P}}_k^{uc})) \tag{6}$$

### 3.3 Overall workflow

Fig. 3 depicts the workflow of Magnet. The target text embedding $\hat{c}$ can be obtained after replacing all object embeddings $c_{E_i}$ with $\hat{c}_{E_i}, i = 1, ..., M$. To generate the image, the pre-trained U-Net denoises the latent $z_{t-1} = \epsilon_\theta(z_t, t, \hat{c})$, where timesteps $t = T, ..., 1$. We set the hyperparameters $\lambda = 0.6$, $K = 5$. Please refer to Appendix C for implementation details.

## 4 Experiments

### 4.1 Datasets

We evaluate our proposed Magnet on two existing benchmarks:

**(1) Attribute Binding Contrast set (ABC-6K) [8].** This dataset consists of natural compositional prompts from MS-COCO [20], each prompt includes at least two concepts (e.g., *"a bathroom with a tan sink and white toilet"*, *"a brown cow standing in a lush green field"*). We randomly sample 600 prompts from this dataset and generate 5 images per prompt to compare all methods.

**(2) Concept Conjunction 500 (CC-500) [8].** The dataset contains prompts that conjunct two concepts, each with one color attribute. Following [7], objects are divided into two types: living (i.e., animals and plants) and other non-living nouns. Prompts type is categorized into (1) two living objects, (2) one living object and one non-living object, and (3) two non-living objects. We adopt 80 prompts for each case to avoid bias and maintain fairness. In total, we have used 240 prompts and generated 10 images for each prompt to compare all methods.

Both datasets are augmented using contrast settings [21]. The position of attribute words for different objects is swapped (e.g., "a red chair and a blue cup" ↔ "a blue chair and a red cup").

### 4.2 Metrics

We mainly rely on human evaluation since the common metrics (e.g. CLIP text-image similarity) are unreliable for assessing attribute binding, which is discussed in Appendix D.

**Coarse-grained comparison**. We assess the generated image for image quality and concept disentanglement on two adopted datasets. To measure ***image quality***, human evaluators were asked "Which image is more realistic or visually appealing?". The evaluation of concept disentanglement is divided into two types: (1) ***object disentanglement*** by asking "Which image shows different objects more clearly?"; (2) ***attribute disentanglement*** by asking "Which image shows different attributes more clearly?". If all images are equally good or bad, evaluators can indicate "no winner". We randomly sample one image for each prompt from ABC-6K and two images for each prompt from CC-500 to conduct this coarse-grained comparison.

Table 1: Coarse-grained comparison on the ABC-6k and CC-500 datasets for image quality, object disentanglement, and attribute disentanglement. Values are normalized to sum to 100.

| | ABC-6K | | | CC-500 | | |
|---|---|---|---|---|---|---|
| | Image Quality | Disentanglement Object | Attribute | Image Quality | Disentanglement Object | Attribute |
| Magnet (Ours) | 26.57 | 25.71 | 27.14 | 25.43 | 24.86 | 29.43 |
| Attend-and-Excite | 15.43 | 21.43 | 19.71 | 22.86 | 26.29 | 18.57 |
| Structure Diffusion | 12.28 | 7.14 | 10.29 | 12.29 | 6.86 | 11.14 |
| Stable Diffusion | 10.29 | 6.57 | 8.57 | 11.14 | 7.71 | 13.42 |
| No Winner | 35.43 | 39.15 | 34.29 | 28.28 | 34.28 | 27.44 |

Table 2: Fine-grained comparison on the CC-500 dataset. For reference, we provide the average confidence (Conf.) of GroundingDINO [22] to detect the object (Det.). Manual evaluation concerns the object existence (Obj.) and the attribute alignment (Attr.).

| | Automatic | | Manual | | Runtime | Memory Usage |
|---|---|---|---|---|---|---|
| Method | Det. | Conf. | Obj. | Attr. | (s) | (GB) |
| Stable Diffusion | 71.5 | 56.4 | 65.8 | 59.1 | 6.62 | 6.1 |
| Structure Diffusion | 72.1 | 56.0 | 64.0 | 63.9 | 7.94 (+20.0%) | 7.0 (+14.7%) |
| Attend-and-Excite | 84.3 | 62.6 | 84.6 | 66.2 | 13.4 (+102.4%) | 15.6 (+155.7%) |
| Magnet (Ours) | 76.5 | 59.8 | 68.6 | 74.0 | 6.81 (+2.9%) | 6.5 (+6.5%) |

**Fine-grained comparison**. This comparison is conducted on the CC-500 dataset based on two key criteria: (1) *object existence*, counting the target objects in the generated images; (2) *attribute alignment*, concerning the correct binding between the object and its attribute. We ask annotators to identify the object mentioned in the prompt per generated image. Take prompt *"a red car and a yellow cat"* as an example, each image will be indicated the number two (show both objects), one (show either *"car"* or *"cat"*), or zero (no distinct object). Attribute alignment is assessed by counting whether the generated object presents the desired attribute (maximum to the number of the generated objects). All generated images on CC-500 are used for this fine-grained comparison. In addition, we adopt the phrase grounding model GrondingDINO [22] to detect the target objects automatically. Note that this automatic detection can not reflect the proper binding.

## 4.3 Quantitative comparison

**Coarse-grained comparison.** In Tab. 1, we present the human evaluation results of Magnet compared to three baseline methods: SD V1.4 [23], Structure Diffusion [8] and Attend-and-Excite [7]. Note that Magnet and Structure Diffusion are both training-free. The ABC-6K benchmark has more complicated and challenging prompts. In this case, all methods may fail to include all objects and attributes, resulting in a higher number of *no winner*. Overall, Magnet achieves the best scores in terms of image quality and attribute disentanglement on both datasets.

**Fine-grained comparison.** As shown in Tab. 2, Magnet alleviates the missing problem more than Structure Diffusion on both automatic and manual evaluation, with 3.8% (Det.) and 4.6% (Obj.) improvement. We are inferior to the optimization method, Attend-and-Excite in object existence. In attribute alignment (Manual Attr.), Magnet outperforms all baseline methods. In addition, we compare the runtime and memory used for generation. The data is obtained by generating 100 prompts each with two images. Obviously, Attend-and-Excite requires more resources which affects efficiency. Conversely, Magnet only adds 2.9% to runtime and 6.5% to memory.

**Evaluation on image quality metric.** We also evaluate Magnet on the commonly used metric FID [24] for two SD versions (V1.4 [23] and V2.1 [25]). We follow the standard evaluation process and generate 10k images from randomly sampled MS-COCO [20] captions. SD V1.4 gets 19.04, with Magnet 18.92; SD V2.1 gets 19.76, with Magnet 19.20, the lower the better. This shows that Magnet will not deteriorate the image quality while improving the text alignment.

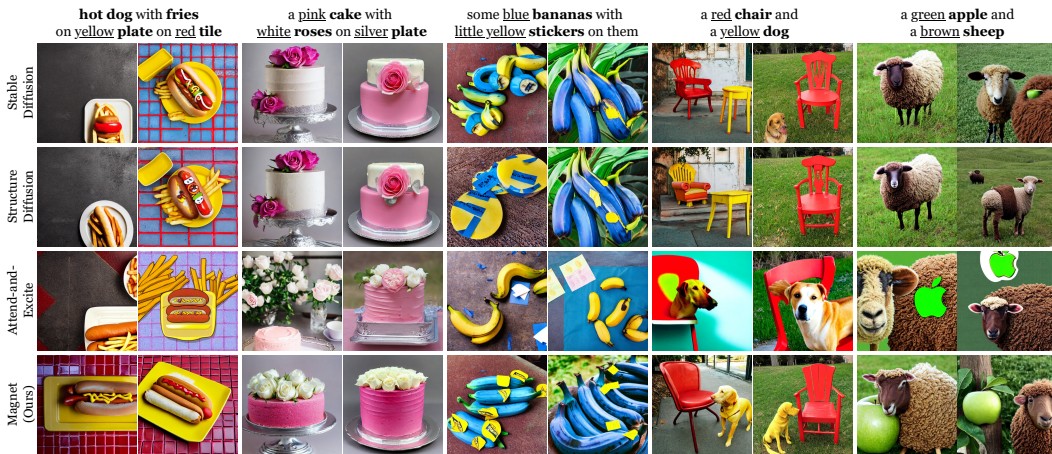

Figure 4: Qualitative comparison using prompts from ABC-6K and CC-500 datasets. For each prompt, we show the image generated by each method under the same seed.

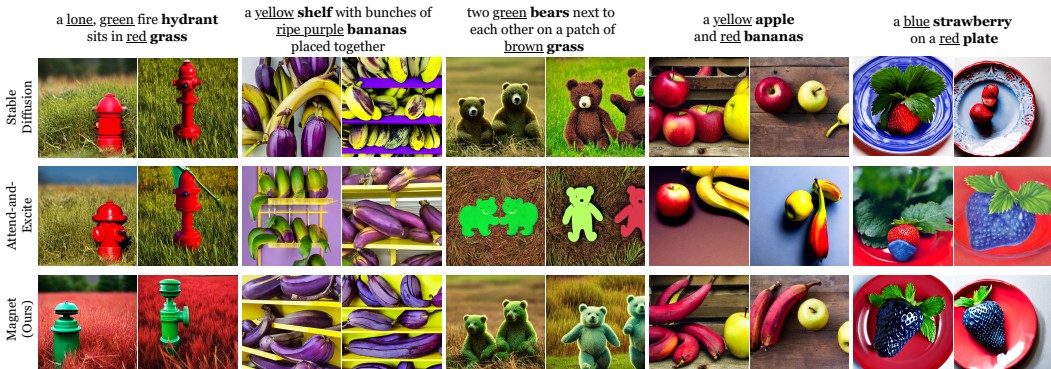

Figure 5: Prompts with unnatural concepts. Baselines generate exchanged colors (row 1) or unwanted artifacts (row 2) while Magnet demonstrates the anti-prior ability with high-quality outputs.

## 4.4 Qualitative comparison

Fig. 4 shows the qualitative comparison of the ABC-6K and CC-500 datasets. The results demonstrate that baselines suffer from the entanglement of objects and attributes.

**Object entanglement** includes the neglect of the object or sticking structures. In columns 1-2, baselines struggle to be faithful to the complex prompt with 4 objects, missing *"fries"* or *"tile"*. In columns 5-6, the objects *"banana"* and *"stickers"* are indistinguishable. Similarly, SD presents blended objects *"dog"* and *"chair"* in columns 7-8 and neglects the target object *"green apple"* in columns 9-10. Note that the results of Structure Diffusion resemble that of SD. On the other hand, the optimization of Attend-and-Excite encourages the attendance of objects but leads to out-of-distribution results, showing strong artifacts (e.g., *"green apple"* in columns 9-10).

**Attribute entanglement** includes the generation of incorrect attributes or the leakage of attributes. For instance, for the prompt *"a pink cake with white roses on silver plate"* with three colors in columns 3-4, SD and Structure Diffusion generate *"white cake"* and *"pink roses"*. In columns 7-8, they generate *"chair"* with mixed colors *"yellow"* and *"red"*. On the other hand, Attend-and-Excite may produce less aesthetic images, which can be attributed to the over-optimized image latent.

Notice that baselines fail to produce **unnatural concepts** like *"blue banana"* in columns 5-6 in Fig. 4. Instead, they generate *"yellow banana"*, which is a natural concept learned as the prior knowledge. Conversely, Magnet is capable of disentangling different concepts and hence generating unnatural concepts, which we call the ***anti-prior*** ability. Fig. 5 displays the results on prompts with anti-prior concepts. We skip Structure Diffusion for its limited improvement over SD.

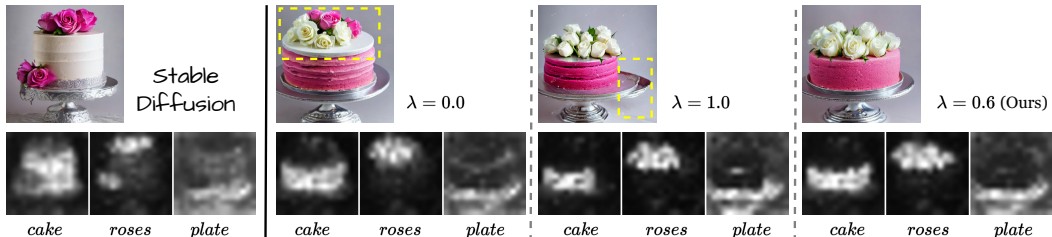

cake    roses    plate      cake    roses    plate    cake    roses    plate    cake    roses    plate

Figure 6: Ablation study on the hyperparameter $\lambda$ given the prompt "a pink cake with white roses on silver plate". A small value of $\lambda$ can not well disentangle different concepts, while a large value causes artifacts in the generated image (best viewed zoomed in). We empirically set $\lambda = 0.6$.

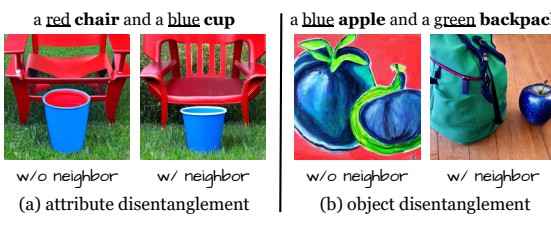

(a) attribute disentanglement     (b) object disentanglement

Figure 7: Ablation study. The neighbor strategy improves the binding vector estimation, separating different attributes (*"cup"* is purely *"blue"*) and objects (*"backpack"* and *"apple"* are distinguishable).

Table 3: Ablation study. Human evaluators were asked to indicate which image can better separate attributes or objects.

|  | Disentanglement | |
| --- | --- | --- |
|  | Object | Attribute |
| w/ neighbor | **27.1** | **28.6** |
| w/o neighbor | 9.1 | 6.4 |
| Stable Diffusion | 2.3 | 2.1 |
| No winner | 61.5 | 62.9 |

## 4.5 Ablation study

**Hyperparameter $\lambda$.** We study the effect of $\lambda$ in Fig. 6. When setting $\lambda = 0$, $\alpha, \beta$ are still positive numbers but the manipulation is in relatively low strength. In this case, concepts are still entangled: *"roses"* appear in shades of *"white"* and *"pink"*. When setting $\lambda = 1$, the result presents artifacts: distorted *"plate"* and watermarked background. We find using $\lambda = 0.6$ can achieve the balance between concept disentanglement and image quality based on the statistic analysis in Fig. 16.

**Selection strategy of the neighbor strategy.** The effectiveness of the neighbor strategy is shown in Fig. 7. The neighbors improve the estimation accuracy and the disentanglement of concepts. In Tab. 3, we ask human evaluators to evaluate both settings using the disentanglement criteria. Evaluators indicate the generated images using the neighbor strategy more disentanglement. This verifies the effectiveness of the neighbor-guided vector estimation.

**Effectiveness of the binding vector.** In Fig. 8, we verify the effectiveness of the binding vector by manually changing $\alpha, \beta$ instead of adaptively calculating by Eq. (3). The value of $\alpha, \beta$ changed from positive to negative shows a swapped binding between objects and attributes. This is because that the context problem in padding embeddings has caused the entanglement of concepts. Our proposed binding vectors can improve the discrimination between objects and lead to designated attributes.

We have conducted additional ablation experiments for the hyperparameter $K$ (Appendix E.1, Fig. 19), and the importance of using both positive and negative binding vectors (Appendix E.2, Fig. 20).

## 4.6 Extensions

**Incorporate with optimization-based methods.** Manipulated in the textual space, Magnet can be readily integrated with Attend-and-Excite. Fig. 9 (a) compares the optimization loss of Attend-and-Excite with and without Magnet. The loss can start at a lower value with Magnet to strengthen the distinction between concepts. Fig. 9 (b) shows vanilla Attend-and-Excite with strong artifacts or inaccurate colors, which should be attributed to the entangled concept representations in padding embeddings. More examples are displayed in Fig. 23 in the Appendix.

**Different text encoders.** In Fig. 10 (a) and (b), we assess Magnet on three T2I models with different text encoders to SD V1.4. Specifically, SD V2.1 [25] adopts CLIP ViT-H/14, SDXL [10] combines

a yellow ($A_1$) **towel** ($E_1$) and a white ($A_2$) **bowl** ($E_2$)

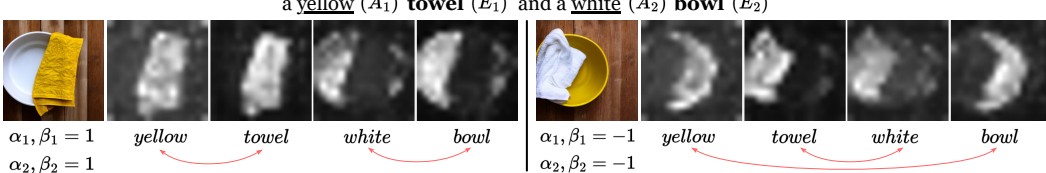

$\alpha_1, \beta_1 = 1$    *yellow*    *towel*    *white*    *bowl*    $\alpha_1, \beta_1 = -1$    *yellow*    *towel*    *white*    *bowl*
$\alpha_2, \beta_2 = 1$    $\alpha_2, \beta_2 = -1$

Figure 8: Ablation study on the effectiveness of the binding vector.

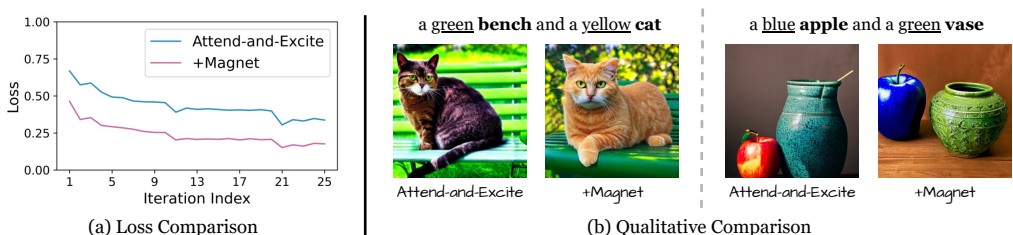

(a) Loss Comparison      (b) Qualitative Comparison

Figure 9: Magnet can be combined with the optimization method, Attend-and-Excite [7]. (a) Magnet improves the loss during optimization. (b) Magnet improves the disentanglement of concepts.

multiple CLIP text encoders, and PixArt [26] uses the T5 encoder [6]. We use the same setting of all hyperparameters and equations for all CLIP-based models while using fixed strength for PixArt. The redesign of the strength formula for the adaptation of T5 is a matter for future work.

**Incorporate with T2I controlling modules.** In Fig. 10 (c) and (d), we investigate the plug-and-play nature of Magnet. Magnet shows compatibility when integrated with existing controlling modules: (1) layout-guidance [27], which constrains the image layout by bounding boxes and intervenes cross-attention layer, and (2) ControlNet [28] conditioned on Depth Map [29] to add spatial control.

**Image editing.** In Fig. 11, we compare the image editing ability of Magnet to Prompt-to-Prompt (P2P) [15], which edits the generated image by manipulating the cross-attention layers. Given the source prompt *"a car on the side of the street"*, we aim to change the attribute of the object *"car"* or *"street"*. In column 1, Magnet applies a positive binding vector $v^{pos}$ (here, the strength $\alpha$ is stated manually) on the word embedding $c_{E_{car}}$ toward the attribute *"old"*. With no control of the attention maps, Magnet surprisingly edits the image with fewer changes in the background than P2P.

# 5 Related work

**Text-to-image diffusion models**. Diffusion models that [30] pioneered, have emerged with great improvement in both unconditional [31, 32] or conditional [28, 33] image generation, together with the advance in synthesis quality [34, 35] and sampling speed [36, 37, 38]. However, the semantic flaw of the text encoder affects the performance of the diffusion models [7, 10, 39]. In this work, we discern the attribute bias and the context issue, providing novel insights about attribute binding.

**Attribute binding**. The binding problem occurs when the model blends improper concepts. To tackle complicated prompts, [9] collaborates different pre-trained diffusion models. [8] suggests word embeddings with blended context and manipulate cross-attention features. In contrast, we highlight the entanglement of the padding embedding and modify solely the text embedding. [7] optimizes the latent to guarantee the attendance of each object. Yet, the optimization may lead to out-of-distribution and require more resources to generate images. Other works [40, 41, 42] introduce layout constraints in the attention layers. Magnet differs from the above approaches in that it can be executed entirely in the textual space. This distinguishes it as a more efficient solution.

It is noteworthy that a line of works [15, 43] achieves image editing on specific visual aspects. However, none have gone as far as this paper in exploring the contextual influences on SD from the perspective of text embedding. Most are subject to a subset of attributes (e.g., texture [44]), control the global object rather than fine-grained attributes [45, 46], or depend on a predefined text pair [47], requiring a learning process or additional datasets. Conversely, our method enhances binding towards arbitrary attributes without the need for new inputs to the standard pipeline.

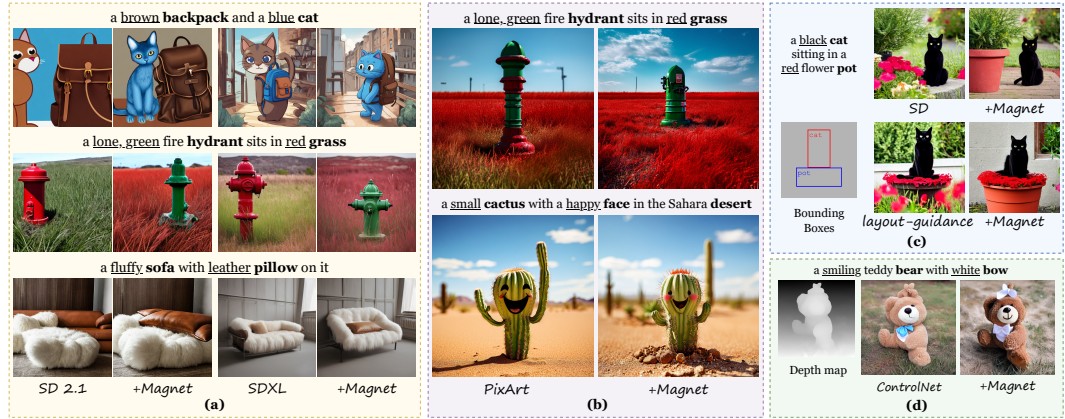

Figure 10: Magnet can be integrated into other T2I models and with existing controlling modules.

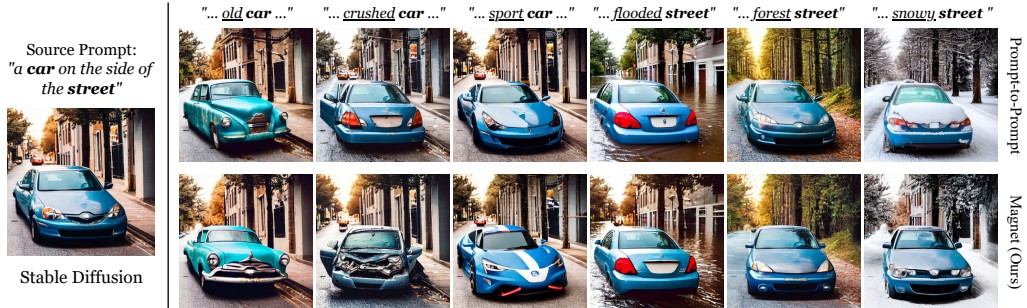

Figure 11: Image editing comparison using prompts from Prompt-to-Prompt [15].

## 6 Limitations

While we have demonstrated improvement in the synthesis quality and text alignment, Magnet is still subject to a few limitations (see Fig. 21). First, it still suffers from the missing problem. In some cases, the manipulation may be overstrength and cause artifacts. An interesting phenomenon is that Magnet generates the correct concepts while rendering errors in positional relations. Finally, it is still challenging to generate an unnatural concept when the object is strongly biased towards one specific attribute. (e.g., *"broccoli"*). We have described the limitations of Magnet in detail in Appendix F.

## 7 Conclusion

In this work, we propose a novel training-free method, Magnet, to tackle the attribute binding issue. First, we conduct a fine-grained analysis of the CLIP text encoder. We observe the phenomenon of attribute bias and point out the context issue of padding embeddings, where the representations of different concepts are entangled, and hence provide potential explanations for existing T2I issues. Second, we introduce the positive and negative binding vectors to enhance the binding within the concept and strengthen the distinction between concepts. Further with the neighbor strategy, the vector estimation can be more accurate. Evaluated in various ways, Magnet shows the ability to disentangle different attributes and generate anti-prior concepts. Performed in the textual space, Magnet improves the synthesis quality and text alignment, with an impressively low increase in computational cost. We sincerely hope that this work will motivate the exploration of generative diffusion models and the discovery of other interesting phenomena.

## Acknowledgements

This work was supported in part by the Natural Science Foundation of China (No. 62272227), and the Postgraduate Research & Practice Innovation Program of NUAA (No. xcxjh20231604).

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

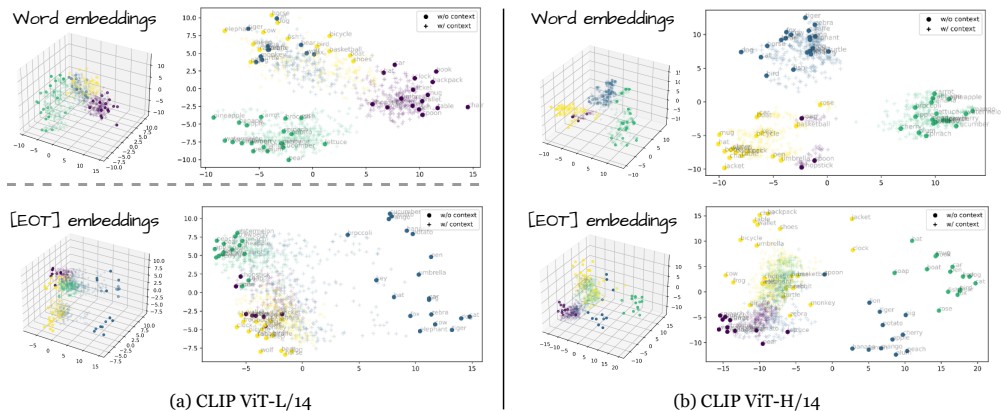

(a) CLIP ViT-L/14          (b) CLIP ViT-H/14

Figure 12: Principal Component Analysis (PCA) analysis of CLIP ViT-H/14 and CLIP ViT-L/14. The word embedding and the [EOT] embedding have a different understanding of the attribute.

# A  Additional analysis of the CLIP text encoder and the diffusion model

## A.1  Analysis of the CLIP text encoder

**Principal Component Analysis (PCA).** We study two types of text embedding through the PCA technique in Fig. 12 for a low-dimensional comparison. We analyze two CLIP text encoders, which are (a) ViT-L/14 (here, the dimension of embedding is $d = 768$), and (b) ViT-H/14 (here, $d = 1024$). We obtain text embedding $c = \{c_{SOT}, c_{object}, c_{EOT}\}$ without the context of the attribute from 60 object nouns (including animals, plants, and non-living entities). We have extended the number of attributes to 16 (including colors and materials), and ended up with 960 text embeddings $c' = \{c'_{SOT}, c'_{attribute}, c'_{object}, c'_{EOT}\}$. We use 60 object embeddings $c_{object}$ or $c_{EOT}$ without the attribute context to fit the model, and then transform contextualized embeddings $c'_{object}$ or $c'_{EOT}$ to the same space. This setting allows us to observe how two types of text embedding understand different attributes. The result indicates that the word and [EOT] embeddings produce different feature spaces with the attribute context. Overall, the distribution of the word embedding is denser, while the [EOT] embedding with attribute context is distributed dispersedly.

**Attribute bias analysis.** Fig. 13 investigates the phenomenon we call attribute bias on two types of text embedding, obtained from the text encoder of CLIP ViT-L/14 and ViT-H/14, respectively. The word embedding without supervision during training has shown severe attribute bias. For example, the word embedding of the object *"tiger"* indicates an extreme preference for the color *"yellow"*. Conversely, the [EOT] embedding produces a relatively small variation in the similarity curve. In the main paper, we conjecture that VLMs' poor compositional understanding and the behavior of bags-of-words [12, 11] on [EOT] lead to an inaccurate textual representation, which affects the interaction between the image latent and semantic word embeddings.

Interestingly, we find that the learned representations of two text encoders are quite different. For example, to encode *"car"* and *"vase"* with the context of different attributes, CLIP ViT-L/14 gets the cosine similarity around $0.6$ v.s. $0.7$, while CLIP ViT-H/14 gets $0.2$ v.s. $0.5$, showing a discrepancy. We conjecture that ViT-L/14 in a large-sized network and dimension may have exacerbated the bias. Yet, it is beyond the scope of our research and we may leave it for future study.

Despite the above difference, both encoders demonstrate the discrepancy between the word and [EOT] embeddings. The stability of the [EOT] embedding can also be explained by entangled context. In contrast, the word embedding without supervision during training may suffer less from entanglement.

## A.2  Fine-grained cases analysis

Take the concept *"red chair"* as an example. The CLIP text encoder maps it into embeddings $c' = \{c'_{SOT}, c'_{red}, c'_{chair}, c'_{EOT}, c'_{pad_1}, ..., c'_{pad_{73}}\}$ (here, 73 for $L - 4$, $L = 77$). The counterpart text embedding of the concept *"chair"* without the color modifier is $c =$

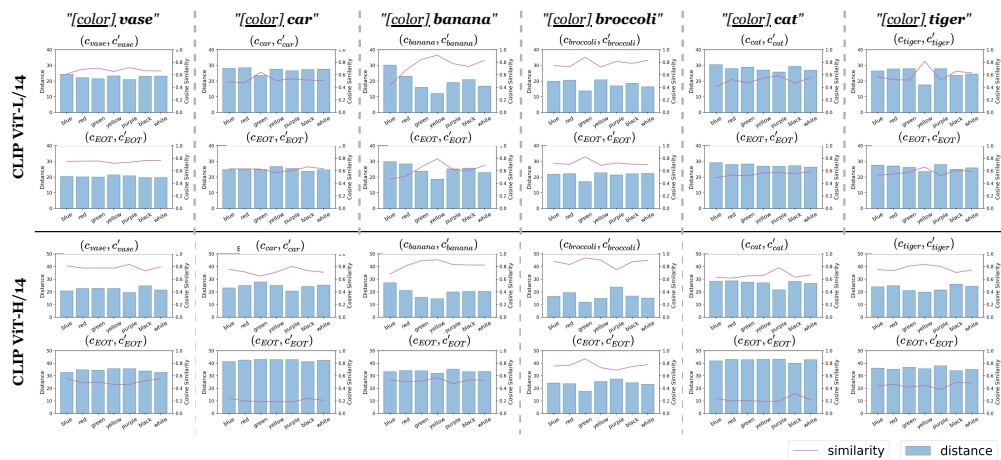

Figure 13: The attribute bias of different objects encoded by CLIP ViT-H/14 and CLIP ViT-L/14. The word and [EOT] embeddings show large discrepancies of attribute bias for the objects *"banana"*, *"broccoli"*, etc. Observe that the extracted embeddings by different text encoders differ significantly.

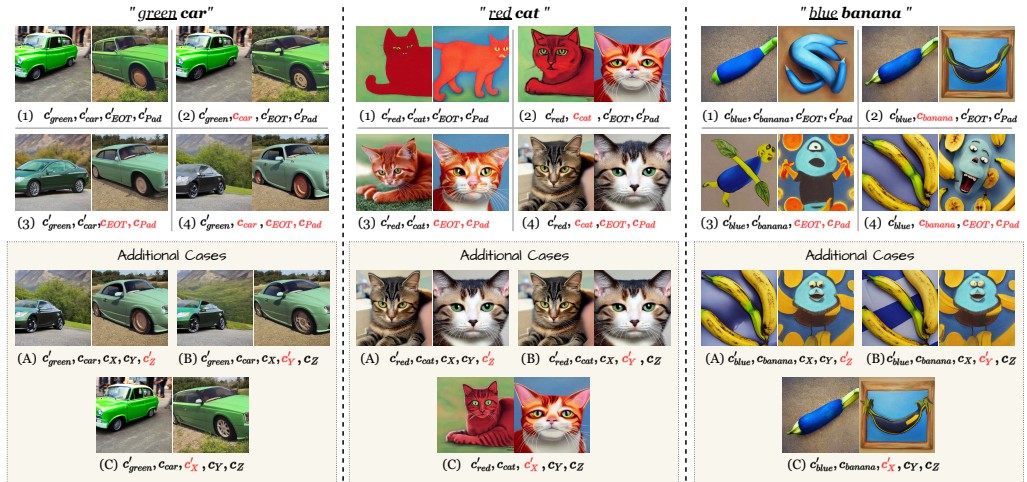

Figure 14: Fine-grained 4 cases described in the main paper, as well as 3 additional cases.

$\{c_{SOT}, c_{chair}, c_{EOT}, c_{pad_1}, ..., c_{pad_{74}}\}$ (here, 74 for $L - 3$). The designed 4 cases are defined as: (1) standard generation conditioned on vanilla embeddings of the concept, i.e., $c_{case1} = c'$; (2) replace the contextualized word embedding in $c'$, i.e., $c_{case2} = \{c'_{SOT}, c'_{red},$ $c_{chair}, c'_{EOT}, c'_{pad_1}, ..., c'_{pad_{73}}\}$; (3) replace all [EOT] and padding embeddings, i.e., $c_{case3} = \{c'_{SOT}, c'_{red}, c'_{chair}, c_{EOT}, c_{pad_1}, ..., c_{pad_{73}}\}$; (4) replace the contextualized word, [EOT] and padding embeddings, i.e., $c_{case4} = \{c'_{SOT}, c'_{red}, c_{chair}, c_{EOT}, c_{pad_1}, ..., c_{pad_{73}}\}$. Note that we maintain the attribute word embedding $c'_{color}$ to observe whether the model can capture the color information without contextual information in other text embeddings. The results are displayed in Fig. 14. As we discussed in the main paper, cases 1-2 where padding embeddings with the color context are still realistic when the concept is natural (i.e., *"green car"*). However, they generate out-of-distribution images for the examples *"red cat"* and *"blue banana"*.

In addition, we have designed 3 new cases to verify that the color information has been gradually *forgotten* in the padding embedding. We divide all [EOT] and padding embeddings into 3 groups: $c_X = \{c_{EOT}, ..., c_{pad_{23}}\}$, $c_Y = \{c_{pad_{24}}, ..., c_{pad_{49}}\}$, $c_Z = \{c_{pad_{50}}, ..., c_{pad_{73}}\}$ (here, these embeddings do not have the color context), and their counterparts $c'_X, c'_Y, c'_Z$ (here, these embeddings with the color context). The results in Fig. 14 (bottom) are consistent with our hypothesis. To be specific, cases A and B show light *"green"* or invisible *"red"* compared to the successful binding results in case C, where embeddings $\{c_{EOT}, ..., c_{pad_{23}}\}$ are contextualized with the target color.

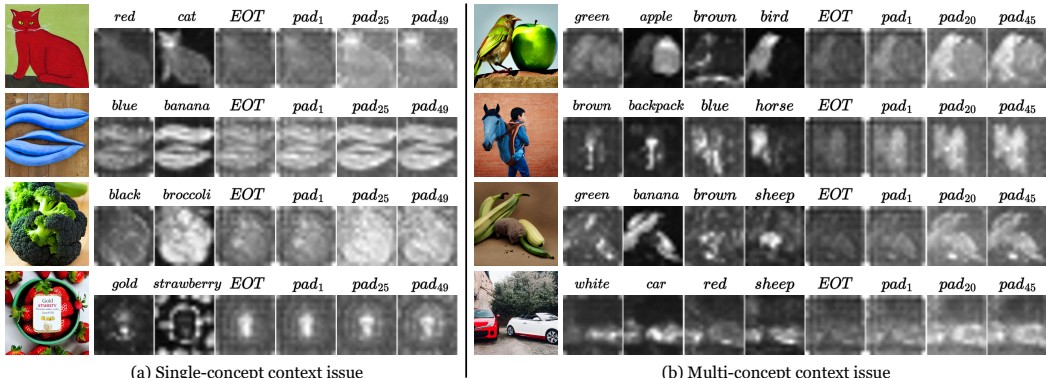

|  | (a) Single-concept context issue | (b) Multi-concept context issue |
| --- | --- | --- |

Figure 15: Several effects of the context issue in padding embeddings under the scenarios of (a) single-concept and (b) multi-concept. We refer to the detailed analysis in Appendix A.3.

**NOTICE**: we propose Magnet based on two key observations on the word embedding. First, the target color is invisible in case 4 for concepts *"green car"*, *"red cat"*. However, these colors can be observed in case 3 (arrive at Eq. (1) for computing $v^{pos}$). Second, cases 1-2 and cases 3-4 for the concept *"blue banana"* both generate catastrophic images. This indicates the vector estimated by the object itself can be inaccurate. In this case, we introduce the neighbor-guided vector estimation.

### A.3 Visualization-based analysis of the context issue

Recall our hypothesis that [EOT] and padding embeddings are trying to *remember* all important information (e.g., attributes, objects, and positions) in the given prompt due to the contrastive learning and bags-of-words behavior of CLIP. In Fig. 15, we investigate the entangled context in the padding embeddings under two scenarios for prompts with a single object or multiple objects.

**Single-concept scenario** aims to generate one object with specific attributes. Fig. 15 (a) shows that the context issue in padding embeddings leads to **(1) out-of-distribution and inaccurate object structures**, e.g., *"cat"* is painting-like in row 1, *"banana"* is unrecognizable in row 2, though presenting correct attributes. Or **(2) generate the object with another attribute** that can compose a natural concept, e.g., *"broccoli"* binds to the prior attribute *"green"* rather than *"black"* in row 3. One potential explanation is the image latent is contaminated by inaccurate representation in padding embeddings, as evidenced by the overlapped activation of latter padding embeddings with the word embedding of each object. The generation of natural concepts proves our hypothesis that latter padding embeddings forget attribute context if the object has a preference for certain attributes based on the training dataset. In row 4, we present an interesting observation that padding embeddings are aligned with the attribute word rather than the object *"strawberry"*. It seems that the word *"gold"* is interpreted as an entity instead of a visual feature, leading to **(3) a split of the target object**.

**Multi-concept scenario** aims to generate multiple objects with the desired attributes. Fig. 15 (b) shows that the context issue in padding embeddings leads to **(4) color leakage**, i.e., one object presents the attribute belonging to another object in row 1. Or **(5) objects stick together**, e.g., a strange creature with the head of a *"horse"* but the body of a *"bag"* in row 2. All the above phenomena can be attributed to the evident entanglement in padding embeddings with overlapped cross-attention activations, which provides inaccurate object representation and indistinguishable binding relationships for each concept. **Note that the above effects can occur simultaneously on a single instance**: row 3 indicates an inaccurate *"sheep"* structure, a binding between *"banana"* and the prior color *"yellow"*, a split of the object *"banana"*, as well as a sticking problem between two objects *"banana"* and *"sheep"*. In row 4, we find that the context issue of padding embeddings also explains **(6) the issue of missing objects**, i.e., the context loses the object "*sheep*" and contains a dominant representation of the object *"car"*.

[18] also discussed the semantic information in padded [EOT] embeddings. While their main concern is to remove one specific **object** content, our focus is the understanding of **attribute**.

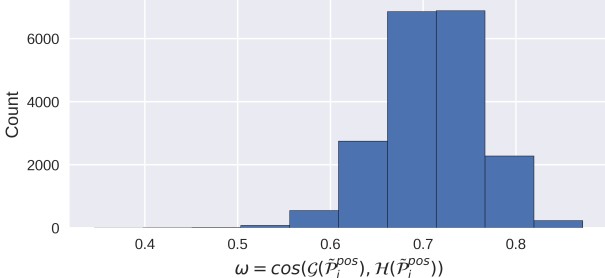

Figure 16: Statistical analysis of $\omega = cos(\mathcal{G}(\tilde{\mathcal{P}}_i^{pos}), \mathcal{H}(\tilde{\mathcal{P}}_i^{pos}))$ obtained from 19648 samples (614 objects and 32 attributes). We set $\lambda = 0.6$ where the count drops.

## B    Detail of the proposed method

### B.1    Dependency parser

To extract the dependency set $\mathcal{D} = \{A_1 \& E_1, ..., A_M \& E_M\}$ in the given prompt, we adopt an off-the-shelf dependency parsing module in Stanza Library [19] and construct syntax trees using NLTK. Following [8], the pair is searched by noun phrases (NPs) in the syntax tree and their corresponding adjective words. For instance, given the prompt *"a black cat sitting in a white bowl"*, the object *"cat"* is extracted according to the label *NN* or *NNs*, then allocated its attribute *"black"* in the subtree. Similarly, the object *"bowl"* and its attribute *"white"* can be obtained. However, the parser may fail to extract the concepts out of the "[attribute] [object]" format. For instance, it can not process the prompt *"a photo of a streetlight that is green"* with dependency *"green"*&*"streetlight"*, or *"apples of green are in white baskets"* with dependency *"green"*&*"apples"*. We leave this for future work.

### B.2    Background of diffusion models

The conventional diffusion model [30] works in two steps: (1) forward diffusion that gradually adds noise to the image $x$; (2) reverse diffusion that removes noise from noisy image $x_t$ step-by-step.

Latent Diffusion Models (LDMs) [2] perform the denoising in the latent space. The pre-trained encoder $\phi$ compresses the image $x$ to the latent $z = \phi(x)$, and the pre-trained decoder $\psi$ reconstructs the latent as $\psi(z) \approx x$. The forward diffusion produces the noisy latent $z_t$ for the step $t = 1, ..., T$. The denoising network $\epsilon_\theta$ is trained to remove the added noise at each step by minimizing $||\epsilon_\theta(z_t, t) - \epsilon||^2$, where $z_t$ is the noisy latent at timestep $t$, $\epsilon \sim \mathcal{N}(0, 1)$ is the added Gaussian noise. The noisy latent $z_T$ is sampled from Gaussian noise $\mathcal{N}(0, 1)$ during inference. Finally, the reversed latent $z_0$ is decoded to produce the image $x = \psi(z_0)$.

The proposed Magnet is applied over Stable Diffusion (SD) conditioned on text prompts. The pre-trained CLIP text encoder $\mathcal{E}$ maps the prompt to the text embedding $c = \mathcal{E}(\mathcal{P})$. SD appends several cross-attention layers to inject the text condition into the latent $z_t$. The loss function of the text-image latent diffusion model can be rewritten as $||\epsilon_\theta(z_t, t, v) - \epsilon||^2$.

### B.3    Strength of the binding vector

The use of the exponential function is inspired by [18]. But in a different way, Eq. (3) that determines the strength $\alpha_i, \beta_i$ is based on our observation in Fig. 2 (b) and (c) in Section 2.

The formula $\omega = cos(\mathcal{G}(\tilde{\mathcal{P}}_i^{pos}), \mathcal{H}(\tilde{\mathcal{P}}_i^{pos}))$ calculates the cosine similarity between the first [EOT] embedding and the last padding embedding of the concept $\tilde{\mathcal{P}}_i^{pos}$. In Fig. 16, we have conducted a statistical analysis using Numpy's histogram to bin the data. Different values $\omega$ are obtained from 19648 samples encoded by CLIP ViT-L/14. The highest counts are at the values 0.66 and 0.71. Observe that the count drops when $\omega < 0.6$ or $\omega > 0.82$. Intuitively, smaller $\omega$ indicates a larger deviation from the target context. Empirically, we set $\lambda = 0.6$ in Eq. (3) to enhance the weak binding (i.e., $\alpha_i > 1$ when $\omega_i < 0.6$ in $e^{\lambda - \omega_i}$). We have conducted an ablation study of the value $\omega$ in Fig.

6. For the strength $\beta_i$ of the negative binding vector, we suggest a relatively slight control to avoid strong deviation when the concept number $M$ is large, i.e., $\beta_i = 1 - \omega^2$.

### B.4 Neighbor-guided vector estimation

**Feature Neighbors.** The candidate set $\mathcal{S} = \{B_1, ..., B_R\}$ used for the feature neighbor strategy includes $R$ words. In practice, we gathered 614 object nouns generated from ChatGPT [13] and checked manually. We extract the word embedding $c_{B_R}$ of each candidate object. For example, the candidate *"truck"* is mapped into $\mathcal{P}(\text{"truck"}) = \{c_{SOT}, c_{truck}, c_{EOT}, ...\}$. The embedding $c_{truck}$ is extracted and used in the formula $d(c_{B_r}, \mathcal{F}(E_i), \tilde{\mathcal{P}}_i^{uc})$. Notice the [EOT] embedding is not used. This procedure of extracting candidates' embeddings is one-for-all, i.e., we compute 641 embeddings once for each new text encoder and save them to the local path.

**Semantic Neighbors.** These neighbor objects are semantically related to the target object. We adopt ChatGPT [13] to predict the semantic neighbors. The instruction follows the sentence pattern of "Which objects are highly related to the word <*> ?". Optionally, the large language model BERT [48] for fill-mask is considered. We mask the object in the prompt to get its neighbors. For example, to predict the neighbor object for *"brown bear"*, the masked prompt is composed as *"brown bear and a [MASK]."*. We hypothesize the conjecture *"and"* can implicitly restrict the close relation. The first two nouns output by BERT are "*wolf*" and "*lion*", which are similar objects to "*bear*".

## C Implementation details

**Configure.** All experiments are conducted on RTX 3090 in a single GPU. Our proposed Magnet is built upon SD V1.4 [23] with the pre-trained text encoder of CLIP ViT-L/14 [5].

**Hyperparameters.** The choice of $\lambda = 0.6$ is explained in Appendix B.3 and verified by the ablation study in Fig. 6. We set $K = 5$ to conduct qualitative and quantitative experiments. We have discussed other choices of $K$ in Appendix E.1. We generate images with 50 diffusion steps with a fixed classifier-free guidance scale of 7.5.

**Baselines.** We compare Magnet to SD V1.4 [2], the training-free method, Structure Diffusion [49], and the optimization method, Attend-and-Excite [7]. Since the official Attend-and-Excite does not provide an automatic parsing process, we extract the required object words (in bold) using the Stanza's package (same to Magnet, see Appendix B.1).

**Datasets.** We have conducted statistics on the CC-500 dataset based on the three types of classification. We find the number of valid prompts for each type are 84, 212, and 136, respectively. This data bias may lead to unfair comparisons. In this case, we randomly select 80 prompts per type and obtain 240 prompts in total.

**Resource.** The runtime to generate an image and the required maximum GPU resources for each method are listed in Tab. 2. The data of each method is obtained by generating 200 images (randomly sampling 50 prompts from each dataset and generating 2 images per prompt). Each method is tested under the same setting to maintain fairness.

## D Metric discussion

We rely on human evaluation since the commonly used metrics for text-to-image synthesis are unreliable for our concern about attribute binding. We discuss three models as the automatic evaluation metrics, which are retrieval models CLIP [5] and BLIP [50], as well as the phrase grounding model GroundingDINO [22].

Fig. 17 (a) shows the drawback of **CLIP score**, which computes the cosine similarity between the text and the image embeddings. Failure and success cases present relatively equal values. The [EOT] embedding suffers from attribute bias and can not measure the unnatural concept *"blue apple"*.

Similar to CLIP, the metric of **BLIP score** in Fig. 17 (b) diverges from the human evaluator. Given the target prompt with multiple concepts, the image of SD (top) presents entangled attributes and objects. In this case, human evaluators indicate no instance of object existence and attribute alignment. However, BLIP text-image similarity can not align with the assessment of human evaluators.

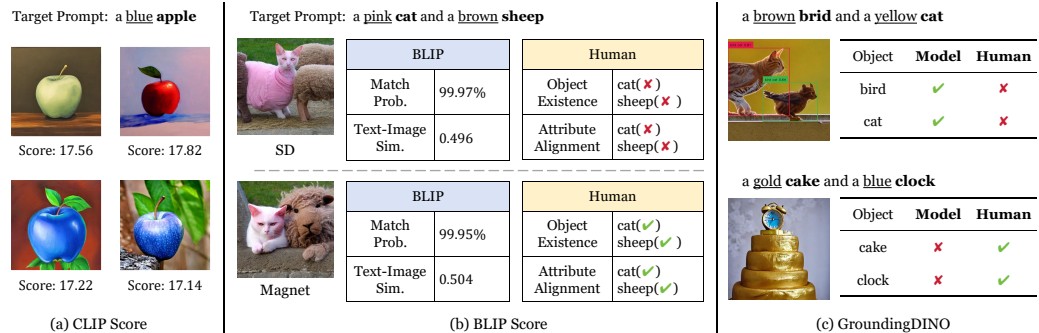

Figure 17: Failure cases of three automatic metrics: (a) CLIP text-image similarity can not assess the binding of unnatural concepts. (b) BLIP text-image similarity fails to capture the entanglement of concepts. (c) The detection of GroundingDINO diverges from human annotators.

Table 4: Quantitative comparison following Attend-and-Excite [7].

| Type | Method | CLIP | | BLIP | |
|------|--------|------|------|------|------|
| | | Full Prompt | Min. Object | Full Prompt | Min. Object |
| | Stable Diffusion | 32.40 | 22.40 | 45.32 | 30.35 |
| Training-free | StructureDiffusion | 32.24 | 22.38 | 44.68 | 30.28 |
| | Magnet(Ours) | 33.11 | 22.79 | 46.53 | 30.84 |
| Optimization | Attend-and-Excite | 34.12 | 24.63 | 49.65 | 34.83 |

In the main paper, we adopt GrondingDINO [22] to detect the object in the generated image. However, it fails to capture the structural deviation and suffers from attribute bias. As shown in Fig. 17 (c), the entangled concepts *"bird"* and *"cat"* are detected by GroundingDINO, which diverges from the human evaluator. Conversely, the model can not detect *"gold cake"*. This may be attributed to the attribute bias, which we have discussed in the main paper.

Additionally, we follow Attend-and-Excite [7] and compare the *full prompt similarity* and *minimum object similarity* using CLIP and BLIP. The quantitative comparison is listed in Tab. 4. Magnet shows improvement on all metrics compared to SD and Structure Diffusion. Meanwhile, we compare the *text-text similarity* [7] using the BLIP model for image captioning, resulting in SD (66.08), Structure Diffusion (65.71), Magnet (68.22), and Attend-and-Excite (71.22) as the highest. However, we do emphasize that **the above quantitative metrics can not reflect the disentanglement of objects and attributes** that we are concerned about.

In conclusion, we refer to the human evaluation to ensure a fair and reliable comparison. A screenshot example of the coarse-grained comparison is given in Fig. 18.

# E   Additional ablation experiments

## E.1   Hyperparameter K

In Fig. 19, we have conducted an ablation study on the hyperparameter $K$ to select neighbor objects. Note that the positive and negative vectors are estimated by each object $E_i$ itself when $K = 1$ as Eq. (1). The difference is slight if concepts in the target prompt are relatively natural. For example, in row 2, using $K = 1, 3, 5$ (column 2-4) can generate the correct concept *"red ball"* compared to *"white ball"* in SD. However, the results of $K = 1$ (column 2) in rows 3-4 present a catastrophic structure of *"blue bananas"*. This verifies the effectiveness of the neighbor strategy. On the other hand, $K$ in a large number can lead to inaccurate binding vectors. For example, in rows 1-2, results of $K \geq 10$ are similar to SD. This can be attributed to the introduction of a multitude of unrelated objects that have an impact on the estimation accuracy. Similarly, *"stickers"* are indistinguishable in row 3, columns 7-8 using $K = 20, 50$.

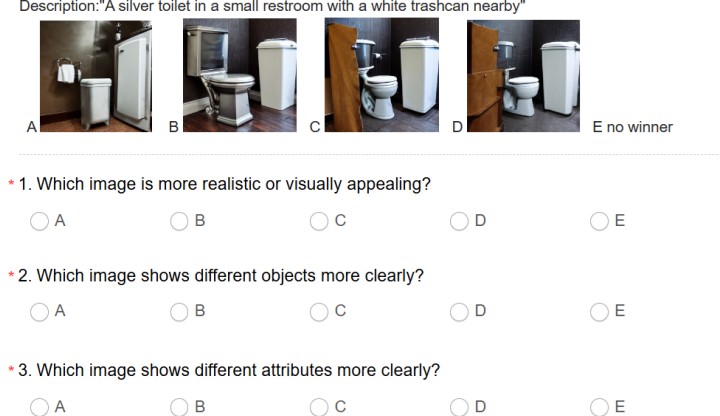

Figure 18: A screenshot of the human evaluation for assessing image quality, disentanglement of objects and attributes. For each question, the order of images generated by Magnet and other methods is randomized to maintain fairness.

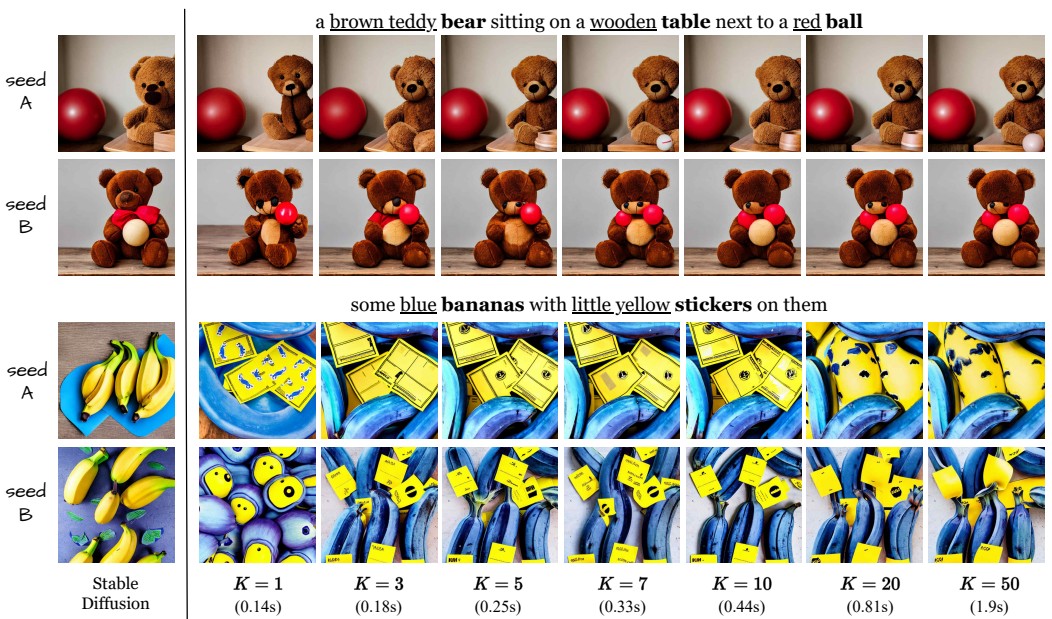

Figure 19: Ablation study on the hyperparameter $K$. We emphasize that $K = 5$ may not always be the best choice because of the randomly initialized latent. For example, the result of $K = 3$ is more appealing than $K = 5$. We choose $K = 5$ which can stabilize the generate of unnatural concepts (e.g., *"blue bananas"* and *"yellow stickers"* can be more distinguishable in $K = 5$ than $K = 3$), as well as balance the processing time.

Interestingly, when using different seeds, the most visually appealing image may not always come from the same $K$. For example, we subjectively prefer the result of $K = 3$ in row 1, but in row 2 the result of $K = 5$ is more appealing. This is due to the randomly initialized latent. Since Magnet's resource requirements are relatively low, we believe it is possible to use different $K$ for the same prompt and generate images simultaneously for freedom of choice to the user.

In conclusion, the reason for the use of $K = 5$ is the balance between synthesis quality and pre-processing time for manipulation. Here, we obtain the data of time by processing 20 prompts, i.e., adding 0.25s to SD to generate an image using $K = 5$. Meanwhile, our code can be improved to shorten the time, which is left for future work.

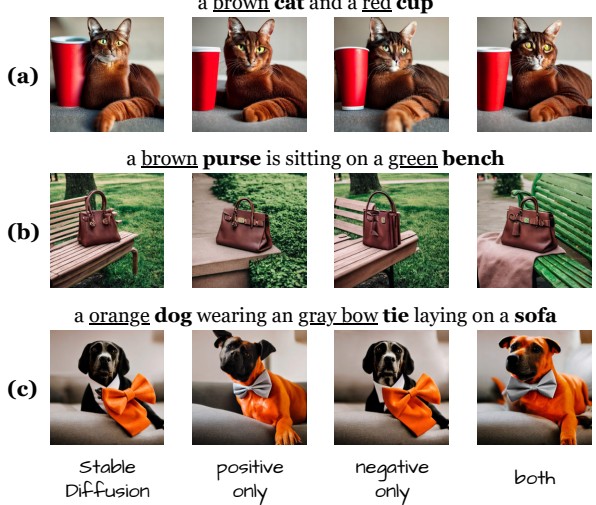

a brown **cat** and a red **cup**

**(a)**

a brown **purse** is sitting on a green **bench**

**(b)**

a orange **dog** wearing an gray bow **tie** laying on a **sofa**

**(c)**

Stable Diffusion | positive only | negative only | both

Figure 20: Ablation study on negative and positive binding vectors. (a) depicts similar results. (b) verifies using both vectors can alleviate the missing object (i.e.,*"green bench"*). (c) verifies using both vectors can enhance the binding (*"orange dog"* and *"gray bow tie"*).

Table 5: Ablation study on negative and positive binding vectors. In most cases, the images generated by three cases are equally good or bad, resulting in a high number of *no winner*.

|  | Disentanglement | |
|  | Object | Attribute |
|---|---|---|
| both | **13.8** | **9.4** |
| positive only | 3.9 | 3.0 |
| negative only | 5.1 | 1.3 |
| Stable Diffusion | 1.4 | 0.2 |
| no winner | 75.8 | 86.1 |

### E.2 Importance of both positive and negative vectors

In Fig. 20 and Tab. 5, we conduct an ablation study on the proposed positive and negative vectors. The object disentanglement is assessed by asking "Which image shows different objects more clearly?", and the attribute disentanglement by asking "Which image shows different attributes more clearly?". We randomly select 12 prompts from CC-500 and 20 prompts from ABC-6K, generating 25 images per prompt (800 images in total) for three settings.

Both qualitative and quantitative comparisons verify the importance of both vectors. For instance, the concept *"green bench"* is interpreted to *"green grass"* when using only one type of the binding vectors. This occurs because of the entanglement of two objects. For the attribute disentanglement, using both vectors is capable of generating objects with desired attributes. Notice that the negative vector improves the object disentanglement (presents *"bench"* in column 5), while the positive vector improves the attribute disentanglement (presents *"orange dog"* in column 3). The human evaluation results are consistent with the above analysis, i.e., negative only (5.1) overpasses positive only (3.9) in terms of object disentanglement, and positive only (3.0) overpasses negative only (1.3) in terms of attribute disentanglement. In conclusion, using both vectors significantly improves text alignment.

## F   Limitations

Although Magnet provides an efficient and effective way to address the attribute binding problem, we acknowledge our technique is subject to a few limitations.

Fig. 21 displays the failure cases of Magnet. First, the neglect of the object (columns 1-2), may be attributed to the model's limited ability to foreground limited subjects. Second, the excessive manipulation of the object embedding leads to out-of-distribution (columns 3-4). An interesting observation is that Magnet sometimes generates images with correct concepts, but incorrect positional relations (columns 5-6). We suspect that the color layout has been determined in the early stage. In this case, Magnet maps the object to the position of the attribute in the image, rather than blending the attribute with the object. Magnet inherits the well-known issue of T2I models, presenting merged objects (columns 7-8). Finally, it is still challenging to generate an unnatural concept when the object has a strong attribute bias (columns 9-10).

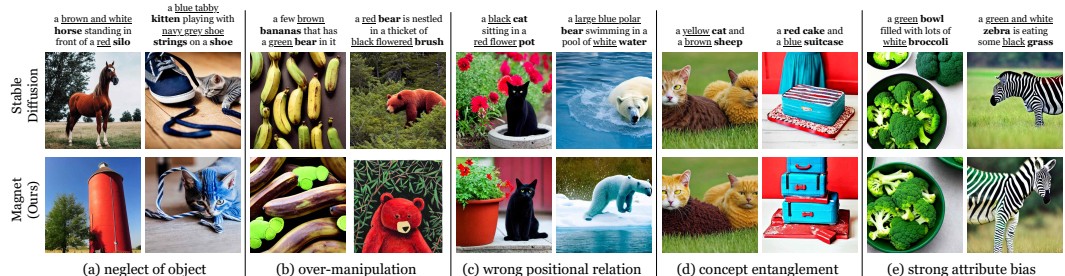

Figure 21: Limitations of the proposed Magnet. (a) shows two cases that amend the concept, while still missing one object; (b) includes out-of-distribution results caused by the excessive value of $\alpha, \beta$; (c) depicts an interesting phenomenon that Magnet correctly disentangles concepts while failing to in accordance with the location word *"in"*. (d) shows Magnet will produce entangled concepts due to the limited power of SD. (e) provides two fail cases to generate unnatural concepts.

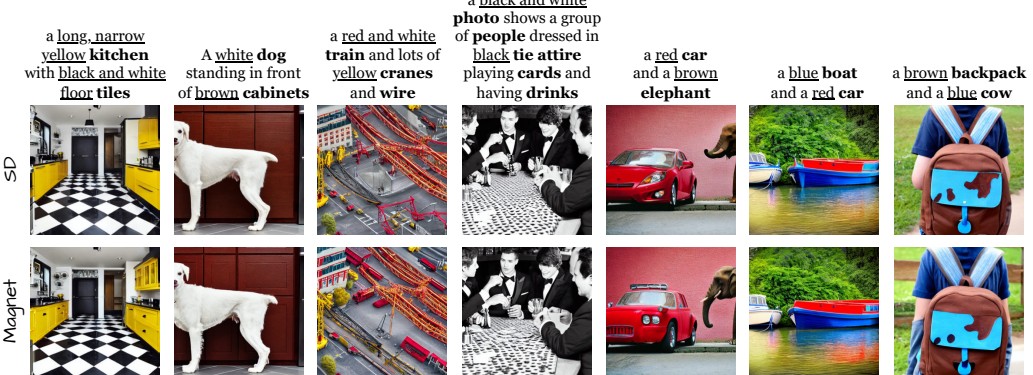

Figure 22: Similar images generated by Magnet and SD.

Additionally, Fig. 22 displays examples that Magnet generates similar images with SD. Most happen when SD has produced relatively faithful images (columns 1-2), or prompts with excessively detailed concepts (columns 3-4), as well as the generation of two unrelated concepts (columns 5-7).

We consider combining Magnet with optimization-based methods to tackle the neglect of objects, e.g., the integration of Attend-and-Excite and Magnet (see Fig. 9 and Fig. 23). Magnet is also compatible with existing T2I controlling modules to address the inability to change spatial relationships, e.g., the integration of ControlNet [28] or layout-guidance [27] and Magnet (see Fig. 10). The excessive or insufficient manipulation may be addressed by improving the formula in Eq. (3), or simply stating the strength $\alpha_i, \beta_i$ manually. We leave these for our future work.

## G   Additional results

Fig. 24 provides examples that Magnet improves the image quality compared to SD.

In Fig. 25, we compare Magnet to SD by visualizing the cross-attention activation.

Fig. 26 provides examples of typical indoor scenes using prompts from the ABC-6K dataset.

Fig. 27 and Fig. 28 provide additional qualitative comparisons on the ABC-6K and CC-500 datasets, respectively.

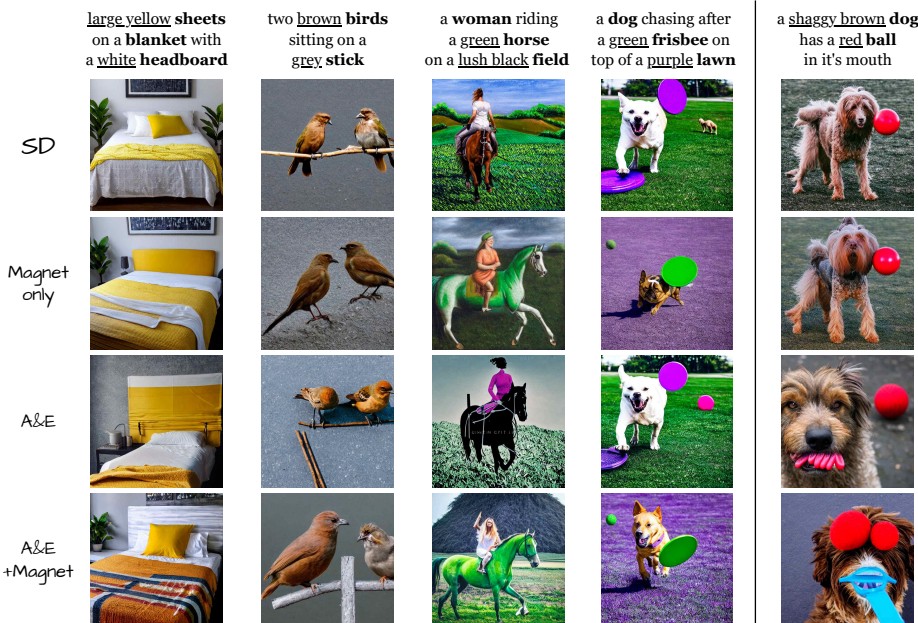

Figure 23: Additional results of extension to Attend-and-Excite. In columns 1-2, Magnet only may neglect the object (e.g., *"gray stick"*). In columns 3-4, Magnet can generate images with unnatural concepts but would be painting-like. The combination (row 4) demonstrates improvement. Column 5 displays a failure case. The parameters may need to be modified to fit Magnet.

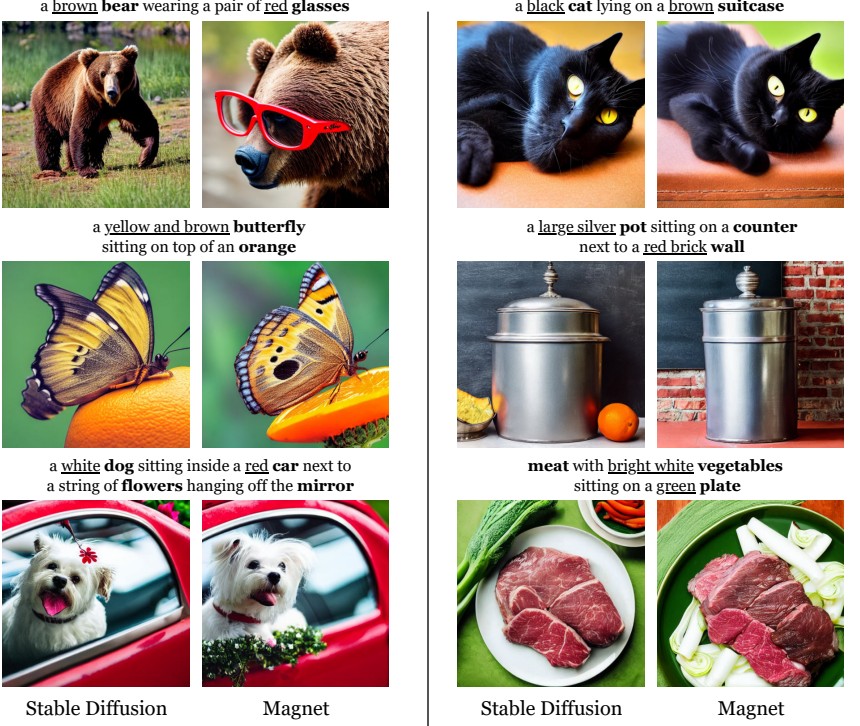

Figure 24: Magnet improves the synthesis quality by disentangling different concepts. Best viewed zoomed in.

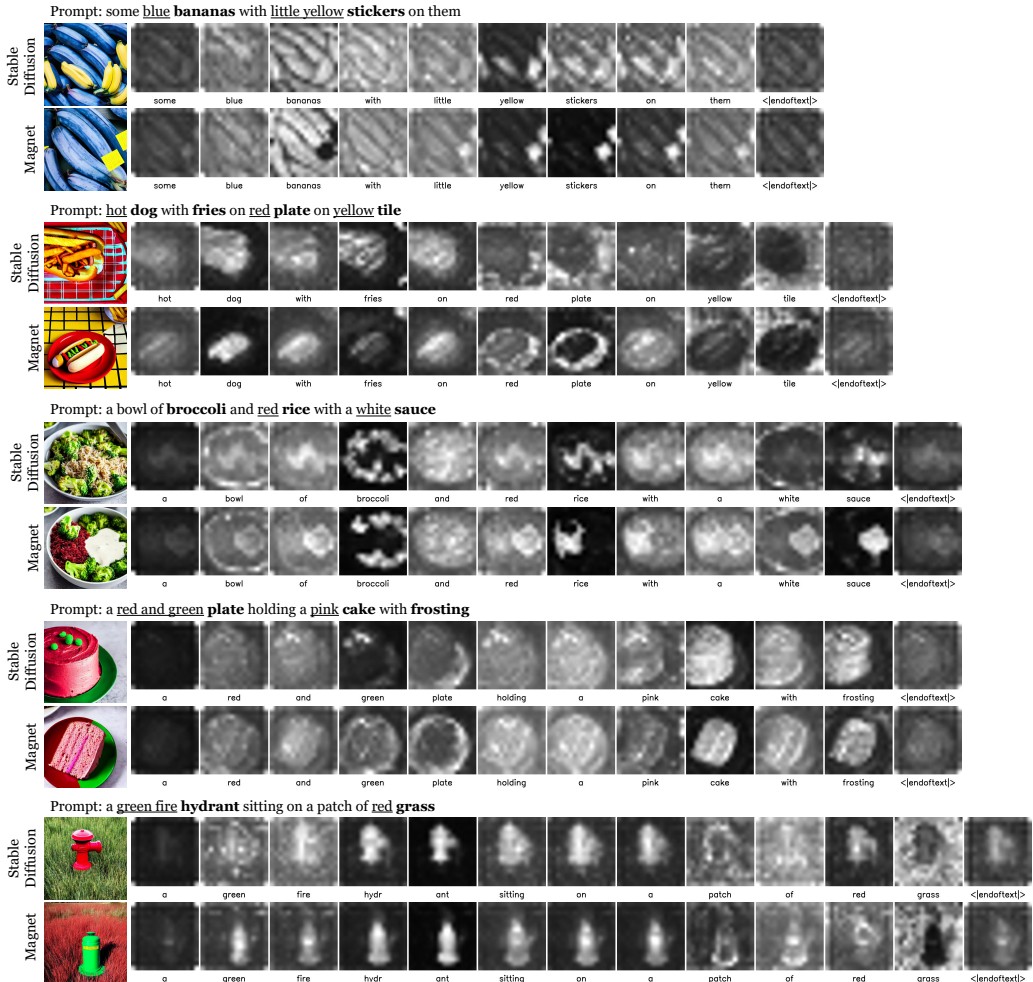

Figure 25: Visualisation of attention maps. The activations of different object are more distinct in Magnet compared to SD. For instance, *bananas* are overlapped with *stickers* in row 1, while row 2 indicates disentangled concepts.

a clean living **area** with a green **sofa** with brown **pillows**

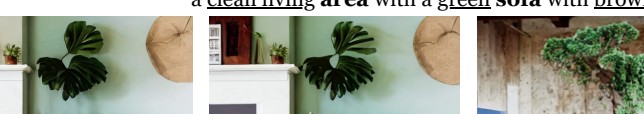
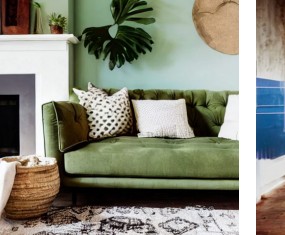
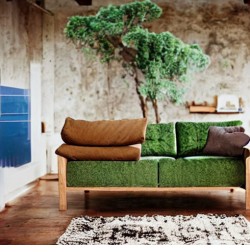
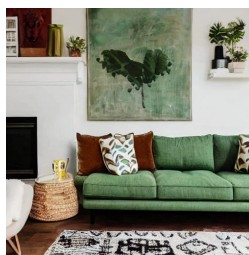

a furnished decorated living **room** with white **walls**, **paintings** and a teal **sofa**

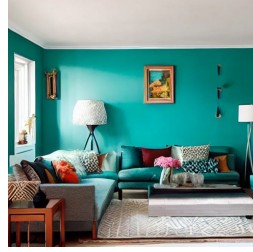
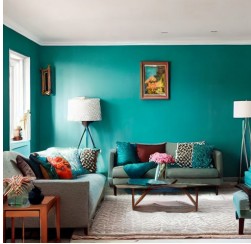
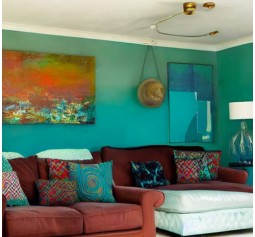
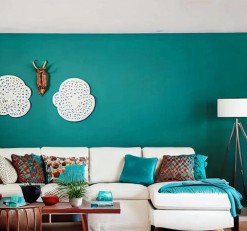

an orange **kitchen** with a white **refrigerator stove oven** and **dishwasher**

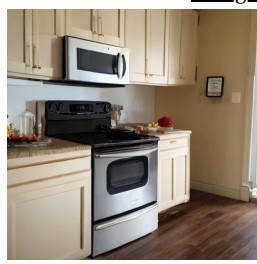
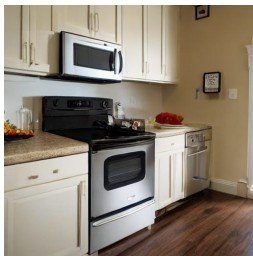
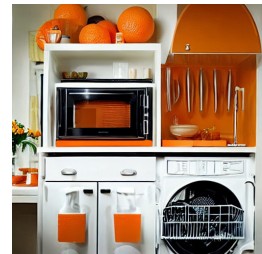
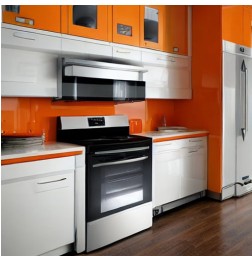

the kitchen with white **oven** atop green tiled **floor**

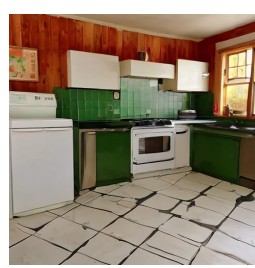
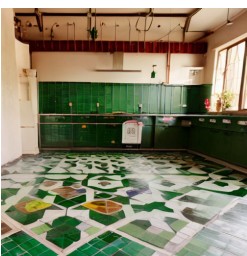
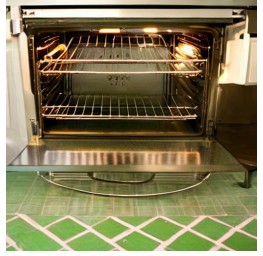
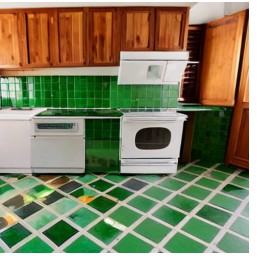

| Stable Diffusion | Structure Diffusion | Attend-and-Excite | Magnet (Ours) |

Figure 26: Qualitative comparison using prompts from the ABC-6K dataset. We provide some typical indoor scene prompts and compare Magnet to baseline methods. Best viewed zoomed in.

a black **train** with three cars is blowing green **smoke**

a red **bench** in front of a stone style **wall** and a **bush** with blue **flowers** to the side of it

a white fire **hydrant** sitting in front of a yellow **fence**

a yellow **teddy bear** in a white **shirt** against a red **wall**

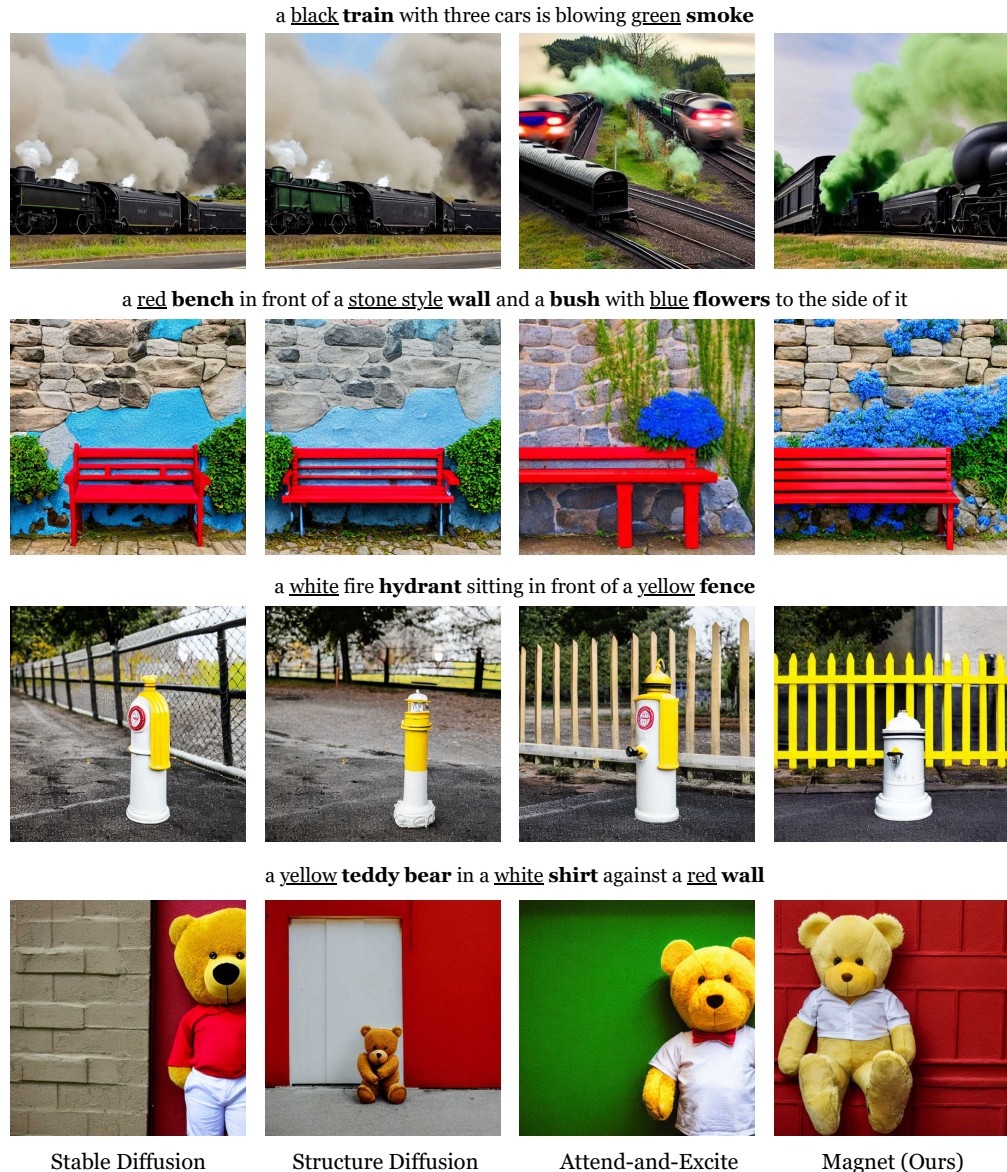

Stable Diffusion          Structure Diffusion          Attend-and-Excite          Magnet (Ours)

Figure 27: Additional results using prompts from the ABC-6K dataset.

a blue **sheep** and a brown **vase**

a gold **car** and a red **clock**

a black **apple** and a green **backpack**

a blue **dog** and a brown **suitcase**

| Stable Diffusion | Structure Diffusion | Attend-and-Excite | Magnet (Ours) |

Figure 28: Additional results using prompts from the CC-500 dataset.

