# OpenReview forum: "Magnet: We Never Know How Text-to-Image Diffusion Models Work, Until We Learn How Vision-Language Models Function"
_NeurIPS.cc/2024/Conference — NeurIPS 2024 poster_

### Official Review · Reviewer_eb4q · 2024-07-11

**Soundness:** 4
**Presentation:** 3
**Contribution:** 3
**Rating:** 6
**Confidence:** 5

**Summary:**

The paper investigates the text embedding representation of text-to-image models in the context of stable diffusion. In particular, the authors find that object features often bind to their commonly associated attributes and propose the Magnet approach that interpolates object features with their designated attributes (positively) and attributes designated with other objects (negatively) specified in the prompt. To determine the strength of the interpolation, the authors leverage the similarity between the EOT and the last PAD token. In addition, for any object word, the authors retrieve their neighboring words based on feature and semantic similarity to further twist the binding vectors, resulting in enhanced concept disentanglement.

**Strengths:**

1. The paper is novel in that it mitigates attribute binding problem from the text-encoder perspective instead of iteratively refining the cross-attention activations that most previous works focused on. This brings both enhanced results and improved efficiency comparatively.
2. The paper's idea is well-motivated with empirical evidence (e.g., using the cosine similarity of PAD and EOT to decide the strength of embedding interpolation by showing that there's more decay in attribute-specific information in later PAD tokens when such attributes are uncommon).
3. The paper is well-written with many design choices are well-justified (e.g., use of human evaluators, strength formula, necessity of both positive and negative binding vectors, etc).

**Weaknesses:**

1. The scope of the work is limited -- it only covers fixing attribute binding problems in text-to-image generation, one downstream task that is commonly encountered in compositional generation [1].
2. The method is model-specific. While it alleviates text-to-image models with a CLIP-based text encoder to a significant degree, it is unknown if this method also improves text-to-image models that use multiple text encoders [2] or different text encoders [3].
3. I find the neighbor finding procedure a bit awkward. For example, the authors have to manually gather 614 object nouns as feature neighbors and have to prompt GPT-3 to gather semantic neighbors, limiting the applicability of the work toward a completely automatic pipeline (e.g., without manual curation or LLM-assisted selection).

[1] T2I-CompBench: A Comprehensive Benchmark for Open-world Compositional Text-to-image Generation
[2] SDXL: Improving Latent Diffusion Models for High-Resolution Image Synthesis
[3] PixArt-α: Fast Training of Diffusion Transformer for Photorealistic Text-to-Image Synthesis

**Questions:**

1. Could authors add reference to Stanza’s dependency parsing module in line 99?
2. Given attribute binding is a core task in compositional text-to-image generation, could authors also discuss more recent and/or relevant in related works, such as [4, 5, 6, 7]?
3. Can authors also evaluate on image quality metrics such as FID [9] and compare with other methods?
4. I wonder if authors could compare Magnet to this concurrent work [8]? As its preprint came out this March, there's no need to experimentally compare and contrast. Some discussions would be appreciated.

[4] Training-Free Layout Control with Cross-Attention Guidance
[5] TokenCompose: Text-to-Image Diffusion with Token-level Supervision
[6] RealCompo: Balancing Realism and Compositionality Improves Text-to-Image Diffusion Models
[7] Compositional Visual Generation with Composable Diffusion Models
[8] Continuous, Subject-Specific Attribute Control in T2I Models by Identifying Semantic Directions
[9] GANs Trained by a Two Time-Scale Update Rule Converge to a Local Nash Equilibrium

**Limitations:**

The paper discussed limitations in missing objects, possibilities of over- and under-manipulation as well as failures in correcting position relationships of the objects with specific examples, which are justifiable given the scope of their method.

---

> ### Author Rebuttal · Authors · 2024-08-07
>
> Firstly, we would like to express our sincere gratitude for reviewing our manuscript and providing valuable feedback. Below are our responses to the weaknesses (W) and Questions (Q).
>
> **W1**: We acknowledge that attribute binding, as our main focus, is part of compositional generation. However, since Magnet performs outside the U-Net, it is plug-and-play for layout-based methods (see Fig. 2(c), layout-guidance+Magnet) or add spatial control (see Fig. 2(d), ControlNet+Magnet) to address the object relationships. Additionally, we have compared Magnet with GORS [1] (see Fig. 2(e) in PDF). Whether trained on color or complex datasets, GORS cannot demonstrate the anti-prior ability and disentangle concepts like Magnet. Meanwhile, it fine-tunes both the CLIP text encoder and the U-Net with LoRA is time-consuming to adapt to different T2I models, while Magnet is plug-and-play. Although the focus of this work is on attribute understanding, we sincerely hope that our work can provide new insights to the community and motivate further work investigating compositional understanding of T2I models.
>
> **W2**: We have applied Magnet to SD 2.1 (another version of the CLIP text encoder), SDXL [2] (multiple text encoders), and PixArt [3] (T5 model) in Fig. 2(a)(b) in the attached PDF. Magnet also improves text alignment and image quality, showing the anti-prior ability. We will add these results to our final version.
>
> **W3**: Magnet is completely automatic during evaluation. Though we describe that we gathered 641 nouns "manually", this procedure is one-for-all. We compute 641 embeddings once for each new text encoder and save them to the local path. In practice, given any prompt, only need to load this local file to search neighbors and encode a few new prompts (e.g., a prompt with 2 concepts, $\sim5\times2\times2$ new prompts are automatically constructed and need to be encoded if $K=5$). This process is fast as shown in Tab. 2 where Magnet has a negligible increase in runtime and memory usage compared to StructureDiffusion and Attend-and-Excite.
>
> **Q1**: Your careful review is much appreciated. We will cite Stanza correctly in the main paper, the same as line 466.
>
> **Q2**: Here is our justification for Magnet different from other works: [4], [6] rely on the layout constraints, which is an additional condition and not included in the vanilla diffusion models, while Magnet only requires the input prompt. We cited two similar works in lines 252-253. [5] controls *objects* in prompts, cannot improve the text alignment for *attributes* like Magnet. [7] composes different noise latents and needs multiple diffusion processes to obtain the target latents. Magnet operates the text embedding outside the U-Net and performs only one diffusion process, which adds negligible cost to the original model and is more efficient. We will discuss these mentioned works in related works in the final version.
>
> **Q3**: We have evaluated on FID for two SD versions (v1.4 and v2.1). We follow the standard evaluation process and generate 10k images from randomly sampled MSCOCO captions. The result is SD v1.4 (19.04), +Magnet (18.92); SD v2.1 (19.76), +Magnet (19.20). This shows that Magnet will not deteriorate the image quality while improving the text alignment.
>
> **Q4**: Thanks for recommending this impressive work. We found some ideas in this paper aligned with ours. But here are some major differences to point out:
>
> 1. It lacks interpretability. In our paper, we analyze the CLIP text encoder and the diffusion model by the 4-case experiment in Fig. 2(a), pointing out the entanglement of the padding embeddings. This has motivated our method and explains why Magnet works (i.e., using the binding vector to enhance the distinction between concepts and improve disentanglement);
>
> 2. It defines "positive and negative prompts" that are semantically contrastive, e.g., "young" vs. "old" for the "age" attribute (may need to manually specify per attribute). In Magnet, positive and negative attributes are obtained from the given prompt automatically, and no need to be semantically opposite. We also introduce unconditional prompts as pivots, which is suitable for attributes that are hard to define their opposite attributes (e.g., what is the contrastive adjective of the color "yellow"?);
>
> 3. Its direction vector is learned with a loss function to ensure robustness. According to the experiment setting, it uses fixed strength (edit delta). Differently, we propose the neighbor strategy and adaptive strength to improve the vector estimation. Encountered with a new attribute, [7] needs a training process to seek this direction, while Magnet can be directly applied.
>
> In the final version, we will discuss this concurrent work [8] in more detail.

---

> > ### Comment · Reviewer_eb4q · 2024-08-07
> > **Response to author's rebuttal**
> >
> > I would like to thank the authors for the rebuttal -- it has addressed all of my concerns to a considerable extent. I'm also impressed that Magnet works just as well on different types of models (e.g., SD 2.1, SDXL, PixArt) and/or for different constrained generation (e.g., Layout-Guidance, ControlNet) based on the provided rebuttal file.
> >
> > Since SDXL and PixArt use different text encoding strategies (e.g., multimodal text encoder or different text encoder), I wonder if the authors could discuss any necessary modifications (e.g., hyperparameters, Magnet formula, etc) to adapt your original setting for the CLIP encoder to work for two encoders (i.e., SDXL) or a T5 encoder (PixArt) to achieve considerable improvement in attribute binding? This would help me better assess the flexibility of your method as well as the scope of impact. Thanks!

---

> > > ### Author Response · Authors · 2024-08-08
> > > **Response to Comment about Flexibility of Magnet**
> > >
> > > Thank you for your thoughtful follow-up and your interest in the flexibility of Magnet.
> > >
> > > When applying Magnet to SDXL, we did not modify any hyperparameters or formulas, keeping all settings consistent with the main paper (e.g., $K=5$, $\lambda=0.6$). The adaptive strength formula (Eq. 3) is designed based on our analysis of the CLIP encoder and naturally adapts to SDXL.
> > >
> > > However, the T5 encoder in PixArt operates differently by modeling bidirectional context. Consequently, it is necessary to conduct further analysis of T5 and redesign the strength formula. Due to the time constraints of the rebuttal process, we were unable to conduct a detailed analysis and therefore used fixed strengths ($\alpha=2$, $\beta=0.5$) for all objects. We also found that $K=10$ is more robust for PixArt.
> > >
> > > We will incorporate these considerations in future work to further demonstrate the flexibility and impact of our method.

---

> > > > ### Comment · Reviewer_eb4q · 2024-08-08
> > > > **Thanks for your response**
> > > >
> > > > Thanks for your response! Given the authors' response, it would be interesting to explore Magnet with a broader scope of text encoding strategies in the future. Regardless, given that mitigating attribute binding issues from the text encoder (as opposed to the diffusion backbone) is under-explored and the authors have done a great job proposing an effective method with some interesting analysis, I would maintain my current rating and confidence and lean toward acceptance.

---

### Official Review · Reviewer_5FTv · 2024-07-12

**Soundness:** 3
**Presentation:** 3
**Contribution:** 3
**Rating:** 5
**Confidence:** 2

**Summary:**

- This work studies how CLIP text embeddings commonly used in text-to-image diffusion models affect attributes in generated images, and how attributes can be bound to the correct objects during generation.
- There is an analysis of the (a) CLIP text encoder and how it interacts with the padding used during T2I diffusion (b) how text embeddings of different nouns relate to different attributes w.r.t to distance (c) how the previous two observations interact during diffusion-based generation.
- An algorithmic innovation — Magnet — is proposed. Magnet introduces a binding vector that can be applied to the embedding of a noun-object to bind an attribute to it so that it is faithfully applied during generation.
- There is a human evaluation (amongst other evaluations) which shows that Magnet improves attribute-object bindings.

**Strengths:**

- The paper is organized very well. I appreciated the bolding.
- The in-depth analysis of what causes the attribute binding problem was interesting and would be widely useful to the community.
- Human evaluation shows that attribute alignment is substantially improved by Magnet. I appreciated the use of a human evaluation rather than merely using automated metrics.
- Magnet is much cheaper w.r.t runtime and memory usage than competing methods.

**Weaknesses:**

I had great difficulty understanding most of the figures. Even after understanding the method, I cannot understand what Fig 3. is showing and how it relates to the method. Similarly, I found the equations and the notations cumbersome, illegible, and frustrating.

I am honestly unsure if I understand the method, because the writing is very unclear (though well organized) and the notation is completely overwrought.

One minor weakness is that the analysis is highly specific to the CLIP text encoder. I don't know how much of a weakness this is practically since the CLIP text encoder is a de-facto standard, but I am writing it here anyway for completeness.

**Questions:**

- Have at least one *simple* figure that explains the *intuition* behind the approach.
- Rewrite 3.1 to be more clear or at least provide examples of all of the $\mathcal{P}$-terms, I am not even sure the notation is correct here.
- More generally, provide a comprehensible explanation of the method so I can confirm my understanding is correct. I'm uncomfortable providing a higher score otherwise, though I would like to, given the amount of work that seems to have gone into the paper.

**Limitations:**

Not applicable.

---

> ### Author Rebuttal · Authors · 2024-08-07
>
> Your positive comment on the contribution of this work is much appreciated. We apologize for any difficulty you may have experienced in following this paper. Hope the following example and the illustration in Fig. 3 in the attached PDF will help you to understand these $\mathcal{P}$-terms and our method.
>
> Consider an input prompt $\mathcal{P}=$"a blue book and a red cup", we use an off-the-shelf parser to extract concepts "blue book"($A_1\\&E_1$) and "red cup"($A_2\\&E_2$). The term $\mathcal{\tilde{P}}$ refers to a new prompt that involves only one object (in line 110, described as "out of the current context of $\mathcal{P}$"). For instance, when operating the object "cup" (i.e., $i=2$), we define positive concept $\mathcal{\tilde{P}}^{pos}_2$="red cup"($A_2\\&E_2$), negative concept $\mathcal{\tilde{P}}^{neg}_2$="blue cup"($A_1\\&E_2$), and unconditional concept $\mathcal{\tilde{P}}^{uc}_2$="cup"($\varnothing\\&E_2$). The function $\mathcal{F}$ extracts the embedding w.r.t. a word in one specific prompt (illustrated by the red box). **Notice that the same word in different prompts with varied contexts will produce different word embeddings**, i.e., $\mathcal{F}(E_2,\mathcal{\tilde{P}}^{pos}_2)\neq \mathcal{F}(E_2,\mathcal{P})$. Based on this fact, we introduce Eq. (1)(2) to estimate the binding vectors $v^{pos}_2,v^{neg}_2$ for "cup". The vector $v^{pos}_2$ identifies the direction towards "red", while $v^{neg}_2$ is the direction towards "blue", specifically for "cup". Similarly, we obtain the binding vectors $v^{pos}_1,v^{neg}_1$ specifically for the object "book"($E_1$).
>
> When involving neighbors, we compute $K$ objects close to the "cup" embedding in the feature space (lines 129-132), denoted $\\{B^{(2)}\_k\\}^{K}\_{k=1}$, where superscript $(2)$ refers to the current object "cup" $i=2$. Next, we replace "cup" in three constructed prompts with each neighbor object. For instance, the second neighbor "mug" ($B^{(2)}_2$, where subscript refers to $r=2$) will produce the positive concept $\mathcal{\tilde{P}}^{pos}_2$="red mug"($A_2\\&B^{(2)}_2$), negative concept $\mathcal{\tilde{P}}^{neg}_2$="blue mug"($A_1\\&B^{(2)}_2$) and unconditional concept $\mathcal{\tilde{P}}^{uc}_2$="mug"($\varnothing\\&B^{(2)}_2$). The same process is for the remaining neighbors to obtain $K$ positive and $K$ negative vectors. The final binding vectors $v^{pos}_2,v^{neg}_2$ used to modify "cup" are averaged as Eq. (5)(6).
>
> The adaptive strength in Eq. (3) improves robustness for practical use. For the object "cup" ($i=2$), we extract the first [EOT], i.e., $\mathcal{G}(\mathcal{\tilde{P}}^{pos}_2)$, and the last padding embedding, i.e., $\mathcal{H}(\mathcal{\tilde{P}}^{pos}_2)$, to compute $\omega_2$ and two strengths $\alpha_2,\beta_2$. Note the used prompt to obtain $\omega_2$ is not the input prompt $\mathcal{P}$ but the constructed positive concept $\mathcal{\tilde{P}}^{pos}_2$. Our motivation for this strategy is described in lines 116-119. Similarly, $\omega_1, \alpha_1,\beta_1$ are calculated for the object "book" ($i=1$).
>
> Finally, we modulate the original object embedding $c\_{E\_i}$ using the adaptive strengths $\alpha_i,\beta_i$ and the estimated positive and negative vectors $v^{pos}_i,v^{neg}_i$, denoted $\hat{c}\_{E\_i}$ as Eq. (4). Note Magnet does not manipulate the cross-attentional activations as existing works do. As described in Section 3.3, Magnet only modifies the embedding of each object word in the input prompt and treats the denoising U-Net as a black box to generate the image.
>
> Magnet provides a simple but effective way to identify the attribute direction for each object, as we said in line 98, to attract the target attribute (i.e., positive vector) and repulse other attributes (i.e., negative vector). Experiments show Magnet disentangling different concepts and improving attribute alignment (see Fig. 4,5 in the main paper).
>
> In Fig. 2(b) in the attached PDF, we show Magnet is not limited to the CLIP text encoder, improving the text alignment and synthesis quality of PixArt [1], which uses T5 as the text encoder.
>
> We sincerely hope that the above explanation will clear up the confusion. We will improve the method section to make this paper more readable.
>
> [1] PixArt-$\alpha$: Fast Training of Diffusion Transformer for Photorealistic Text-to-Image Synthesis

---

> > ### Comment · Reviewer_5FTv · 2024-08-13
> >
> > Thank you for your effort. Unfortunately, your explanation in the comments did not help. I strongly encourage you to choose different notation. I will keep my rating; I think there is a clear technical contribution in this paper, but the comment did not clear anything up for me. I have put my confidence at a 2; the AC can disregard my review at their discretion.

---

### Official Review · Reviewer_iZtx · 2024-07-13

**Soundness:** 4
**Presentation:** 3
**Contribution:** 2
**Rating:** 6
**Confidence:** 3

**Summary:**

The authors propose _Magnet_ to solve the attribute binding problem. (1) Initially, they specifically analyze how the improper binding problem occurs in text embeddings. By comparing embeddings for each token, they demonstrate the attribute bias phenomenon where attributes do not bind well to the object token $c_{object}$ in rare concepts like "blue apple." Additionally, through cosine similarity analysis of padding embeddings, they hypothesize that the latter padding tokens, e.g., $pad_{73}$, tend to represent prior bindings (like red apple) rather than binding the target concept (blue) with the object (apple). (2) Magnet suggests a method to slightly edit the embedding of the object to align with the desired concept, based on these observations. They demonstrate their method's efficacy through ARC-6k and CC-500 datasets.

**Strengths:**

- Writing quality

The writing quality is good for understanding the core motivation and method. Analyzing embeddings using cosine similarity is intuitive and effectively demonstrates how attribute bias negatively impacts attribute binding. The flow from the initial analysis to the method is also easy to follow.

- Novelty

Resolving attribute binding in the text space, as done by Magnet, seems quite plausible and novel. Identifying the limitations of the T2I diffusion model from the VLM model has been necessary but has not been extensively conducted until now.

**Weaknesses:**

Overall, I think it's a good paper. It would be better if the evaluation were a bit more solid.

- Evaluation

I agree that a user study w.r.t performance comparison as shown in Table 1 is necessary, but for this task, it is necessary to evaluate whether each method achieved binding rather than comparing different methods. Since we don't know the precise performance of each technique, it's difficult to understand the actual performance of the Magnet and how much it has improved.

The results in Table 2 are weak. It's disappointing that the automatic results are quantitatively worse compared to attend-and-excite. While I agree that manual inspection is the most accurate, trying out a recent VLM like GPT-4o, which can discern characteristics that image encoder-based evaluations fail to capture, might be worthwhile.

**Questions:**

Q1: How is the analysis in lines 64-78 and Figure 2-(a) connected to the method?

Q2: Is there no need to modify the padding? If the padding tokens positioned at the end contain prior knowledge, it seems they might also interfere with binding.

Q3: How did you perform detection using GrondingDINO in Table 2? I am curious whether you detected "blue apple" or just "apple." It might be somewhat out-of-scope, but I am also interested in how different the results are between detecting "blue apple" and "apple."

Q4: As mentioned in the limitations (Figure 20), Magnet seems relatively weak at handling positional information. I suspect this issue arises not so much from a problem with Magnet itself but rather because CLIP's embedding does not effectively bind positional information to the desired token. Is there an analysis similar to Figure 1 for positional information? It seems only color is present in the appendix. Showing this could further demonstrate, as per the title of your paper, an understanding of VLM’s function, providing clarity on what can and cannot be bound.

---

- Relevant references

Before this paper was submitted, methods for editing text tokens were proposed in papers like [1, 2]. Although these papers also deal with a different task, and attribute binding, and were not considered in the evaluation due to being on Arxiv, I recommend citing them in the camera-ready version.

[1]: Uncovering the Text Embedding in Text-to-Image Diffusion Models, https://arxiv.org/abs/2404.01154

[2]: TexSliders: Diffusion-Based Texture Editing in CLIP Space, https://arxiv.org/abs/2405.00672

**Limitations:**

The authors describe the failure cases of Magnet in the appendix. They have adequately addressed the limitations.

---

> ### Author Rebuttal · Authors · 2024-08-07
>
> We sincerely appreciate your thorough review of our manuscript and the insightful comments provided. Here are our detailed responses to the identified Weaknesses (W) and Questions (Q).
>
> **W1**: Our comparison experiment has been designed to evaluate whether each method achieved binding. In Tab. 1, *attribute disentanglement* is assessed by asking "Which image shows different attributes more clearly?". Only the method achieving the best binding will be voted, otherwise evaluators tend to choose *no winner*. In Tab. 2 *attribute alignment* (Attr.), human annotators annotate a precise number of successful bindings (whether one object presents the desired attribute). This is done separately for each method, without comparing the others. Lines 161-178 have explained each criterion in detail.
>
> **W2**: Automatic results in Tab. 2 only assess *object existence* (see Q3). Magnet is inferior to Attend-and-Excite (Attend) since it optimizes the noisy latent to encourage the model to attend to all objects. As discussed in limitations, we acknowledge Magnet suffering from missing objects. Though designed to enhance *attribute alignment* (has outperformed all baselines in Tab. 2), Magnet shows improvement in *object existence* (Det. $+5$ on SD), which is a bonus of disentangling different concepts (see Fig. 6(b)). This still holds when integrating Magnet with Attend. We tested Attend+Magnet on CC-500 via GroundingDINO. The Det.($\uparrow$) result is 87.7, compared to 84.3 w.r.t the original Attend. Attend+Magnet can encourage all objects and improve attributes simultaneously (see Fig. 8). Since there is no available API at this moment, we cannot evaluate Magnet with GPT-4o. But we will definitely try out GPT-4o whenever possible.
>
> **Q1**: The 4-case experiment is deeply connected to our method. Lines 64-78 and Fig. 2(a) focus on comparing how the context information in the word embedding affects generation (i.e., from case 1 to 2, or case 3 to 4), and how paddings' context information affects (i.e., from case 1 to 3, or case 2 to 4). Appendix A.2 provides a detailed analysis of these fine-grained cases. We discern that the attribute information is rich in word embeddings but entangled in padding embeddings, and modifying these word embeddings will not change the image layout as significantly as modifying padding embeddings. These have motivated us to introduce the binding vector, adaptive strength, and neighbor strategy (see lines 439-443).
>
> **Q2**: We did consider modifying the padding tokens, but for the following reasons it didn't work:
> 1. These padding embeddings strongly entangle different concepts, especially given complex prompts. It is hard to manipulate one specific object from these paddings;
> 2. As shown in Fig. 2 and Fig. 12 in the main paper, modifying the padding (from cases 1 to 3) will change the image significantly compared to modifying the word embedding (from cases 1 to 2). We consider not changing the non-attribute part can better maintain the pre-trained model's capability;
> 3. As shown in Fig. 1 in the attached PDF, the padding embeddings are less activated than word embeddings, especially in U-Net's last two upsampling blocks and the later diffusion steps, which are more crucial for generating semantic features.
>
> **Q3**: In our comparison experiment, we input all object words in the prompt to GroundingDINO. For example, given the prompt "a blue apple and a green vase" with two objects, GroundingDINO's input is *"apple . vase ."* without any adjective. We have considered inputting adjectives, i.e., *"blue apple . green vase ."*. However, no matter what color the apple is (e.g., blue, red, or green), GroundingDINO in both settings will detect it. That's why we use GroundingDINO to assess *object existence* only, but not *attribute alignment* (see lines 177-178). Appendix Fig. 15(c) presents GroundingDINO's failure cases. We further conducted the following experiment to make our judgment sound. Given 50 images (generated by the prompt "a blue apple and a green vase"), GroundingDINO detected $38$ (w/o adjective), $38$ (w/ adjective), human annotated $\sim 20$ "blue apple", maximum is 1 per image. This comparison shows that GroundingDINO is not reliable for measuring *attribute alignment*.
>
> **Q4**: We agree with your idea that the CLIP's embedding does not effectively bind positional information to the desired token. However, this makes an analysis like Fig. 1 difficult - we don't know which token to identify. We conjecture that the positional information is used in the early diffusion steps, where all tokens have relatively small attention values (see Fig. 1 in the attached PDF). In this case, the entangled padding embeddings may affect generation. This also explains why Magnet modifying the word embeddings cannot address the positional problem. However, we can integrate Magnet with layout-based methods to handle position (see Fig. 2(c) in the PDF). Except for colors, Magnet is capable of binding other attributes (see Fig. 2(a) row 3 and 2(b) row 2 in the PDF). The final version will include a detailed discussion of positional relationships and more examples.
>
> Finally, the two textual-based image editing works are impressive. We will cite the given papers and discuss the differences in the final version.
>
> [1] T2I-CompBench: A Comprehensive Benchmark for Open-world Compositional Text-to-image Generation

---

> ### Comment · Reviewer_iZtx · 2024-08-13
>
> Thank you to the authors for their detailed response. I have carefully considered the points raised.
>
> > [W1] "Our comparison experiment has been designed to evaluate whether each method achieved binding. In Table 1, attribute disentanglement is assessed by asking, 'Which image shows different attributes more clearly?' Only the method achieving the best binding will be voted for; otherwise, evaluators tend to choose no winner."
>
> While I understand the authors' approach, I still believe that this method inherently involves a comparison. Evaluating A vs. B seems likely to yield different insights than evaluating A and B separately. The authors suggest that both approaches could lead to similar results, but I respectfully disagree, as I believe pairwise comparisons and individual method evaluations may not produce the same outcomes.
>
> > [W2] "Since there is no available API at this moment, we cannot evaluate Magnet with GPT-4o. But we will definitely try out GPT-4o whenever possible."
>
> I appreciate the authors' intent to evaluate with GPT-4o when possible. However, isn't the GPT-4o API already available?
>
> > [Q3] "This comparison shows that GroundingDINO is not reliable for measuring attribute alignment."
>
> I understand this concern, and I recognize the challenges it presents. It is unfortunate that there currently isn't a more robust evaluation method beyond manual inspection. I had hoped that GPT-4o might offer some supplementary capabilities in this area.
>
> Given these unresolved concerns regarding the evaluation, I will maintain my score as a weak accept.

---

### Official Review · Reviewer_2c6i · 2024-07-15

**Soundness:** 2
**Presentation:** 2
**Contribution:** 4
**Rating:** 6
**Confidence:** 4

**Summary:**

The paper analyzes and improves upon the “(attribute) binding problem” in VLMs such as CLIP, and the focus is primarily on the text encoder side.
First the authors analyze how the individual text embeddings behave in a diagnostic setting when encoding a two-word text “COLOR OBJECT”. With these insights they propose to modify the original embedding of an object by calculating a “binding vector” that uses a positive attribute (the one actually in the prompt, i.e. “yellow”) and negative attributes (other colors).
They show improvements to previous methods on top of which they build such as Structure Diffusion.

**Strengths:**

The paper is very practical and intuitive, motivating the final method with analysis of the embedding space. It has a clear contribution and the whole story ties around this contribution, providing additional insights.

On top, the evaluation is thorough and uses human judgment. (there is no automatic metric here, I agree).
It is also good to see that the neighbor strategy was quantitatively ablated with human judgment, and section 4.6 adds additional nice insights, albeit not beyond a few examples as far as I can tell.

Overall a very interesting and creative exploration, and the empirical results do show that the method works! That is exciting, given that this is a persistent major problem many papers have addressed.

**Weaknesses:**

While the method clearly improves upon previous work, it requires pre-computing or manually defining a lot of components such as the negative attribute set or the neighbor set, with additional computation of the adaptive strength. With pre-computation this is not too expensive but it feels somewhat hacked together.

The abstract mentions too many terms where it is unclear whether they are established terms and if not, the abstract is not the right place to mention them all: blended text embeddings, attribute bias, object embedding, binding vector.

Since this papers main focus is a thorough analysis and interpretability insights, it has to be clearer with its definition, i.e.:
“The above phenomenon, which we call attribute bias, occurs when one object prefers or dislikes
a specific attribute” → refer/dislike are not well-defined.

The 4 experiments with swapping various embeddings seem very interesting but it is not motivated why we need all 4, maybe 2 cases would have been more insightful and easier for the reader to understand what’s going on?

Finally, having more quantitative statistically significant experiments and less showing of examples would have strengthened the paper's contributions further!

**Questions:**

It says in the abstract “blended text embeddings as the cause of improper binding… “. Is “blended text embedding” embeddings a term people are expected to know? I work in vision-and-language and have not heard of it.

Again in abstract “we observe the attribute bias phenomenon”: was this meant to say attribute binding?

How are negative attributes exactly determined? Most things are not as easy as color.

Is there an explanation why the Structure Diffusion baseline is so bad?
Does your method differ to Stucture Diffusion only in the proposed Magnet method and nothing else? Same exact SD version, CLIP version etc.? This is important to clarify in the paper so that it is clear that your method alone leads to this improvement.

Note: Fig. 1b is very small and also low-resolution, please fix!

**Limitations:**

Yes, addressed.

---

> ### Author Rebuttal · Authors · 2024-08-07
>
> We sincerely appreciate your careful review and the valuable suggestions provided. Our responses to the Weaknesses (W) and Questions (Q) are outlined below.
>
> **W1**: Magnet is fully automatic during evaluation and does not require manual definition of these positive/negative attributes. We have introduced these components to ensure robustness. Various ablation experiments have proved the effectiveness of each component, including the parameter $\lambda$ (see Appendix Fig. 17), the neighbor strategy (see Fig. 18), and the positive and negative vectors (see Fig. 19). Meanwhile, our method can be simplified by using fixed strengths and estimating the binding vector without neighbors (i.e., by the object itself). Furthermore, as you pointed out, it is not expensive at all.
>
> **W2&Q1&Q2**: Your comments on our abstract are very valuable to us. The following is our explanation of the terms used in the abstract:
> 1. the term "blended text embeddings" in other words means "text embeddings are blended", with reference to [1] "tokens in the later part of a sequence are *blended* with the token semantics before them";
> 2. the term "attribute bias phenomenon" appears in line 60, i.e., our discovery based on the analysis of the CLIP text encoder, rather than the "attribute binding problem".
>
> We will check all the terms in the abstract, and improve it to be clear and professional.
>
> **W3**: This sentence can be explained with an example in Fig. 1(b) (apology for its blur, we have provided a high-resolution one in Fig. 1 in the attached PDF). For the object "banana", both word and EOT embeddings show high similarity on the color "yellow" but low similarity on "blue", as if "banana" *prefers* the attribute "yellow" but *dislikes* "blue". We will improve the analysis section and clarify all definitions in the final version.
>
> **W4**: The 4-case experiment is deeply connected to our method. As discussed in lines 439-443, our motivation is based on several observations between 4 cases. Specifically, case 1 is a reference case w.r.t. standard generation, case 2 shows how the context information of word embeddings affects generation, case 3 shows how padding embeddings' context information affects generation, and case 4 tests whether the model can capture attribute information in adjectives (described in lines 428-429). Using all 4 cases would be more convincing to introduce our proposed binding vector and adaptive strength. In addition, we add 3 new cases in the Appendix to justify the information-forgotten problem in the latter padding embeddings (see Fig. 12, cases A-C). We will add more descriptions of this experiment to the main paper and improve readability.
>
> **Q3**: We define "negative attributes" as adjectives in the given prompt that do not belong to the current object. For instance, given the prompt "a green apple on a wooden table and a red chair" with three concepts, when operating the object "apple", its positive attribute is "green", then negative attributes are "wooden" and "red" (belong to "table" and "chair", respectively). It should be noted that negative and positive attributes do not need to be semantically opposed to each other. Magnet is not limited to colors and can work on prompts with extensive attributes (see Fig. 2(a) row 3 and (b) row 2 in the attached PDF). We will add more non-color examples in the final version.
>
> **Q4**: We assure that our experiments are completely fair through consistent control of all settings (e.g., the same SD&CLIP version and seeds). Its poor performance can be explained by our discovery in the text encoder: the improper binding is caused not only by the word embeddings, but also by the entangled padding embeddings. StructureDiffusion simply separating the word embeddings of different concepts is insufficient to offset the entanglement in the padding embeddings. Conversely, we introduce binding vectors to reinforce the difference between each concept. This improves disentanglement. Additionally, StructureDiffusion also performed poorly in [2]'s comparison experiments - Structure Diffusion may even get worse scores than Stable Diffusion.
>
> Finally, we sincerely apologize for Fig. 1 and will fix it in the final version. Your feedback is very helpful to us. We will improve our paper to strengthen our contribution, reducing visual examples and adding quantitative results (e.g., FID-10k: SD 19.04, Magnet 18.92).
>
> [1] Training-Free Structured Diffusion Guidance for Compositional Text-to-Image Synthesis
>
> [2] Attend-and-Excite: Attention-Based Semantic Guidance for Text-to-Image Diffusion Models

---

> > ### Comment · Reviewer_2c6i · 2024-08-08
> >
> > Thank you for the explanations and the new PDF! It is good to see you provided new results and engaged with all the questions. I still think my score of 6 is appropriate and think this paper is a weak accept.

---

### Author Rebuttal · Authors · 2024-08-07

In the attached PDF file, we provide a new perspective on how the word and padding embeddings affect generation (see Fig. 1), additional examples applying Magnet to different T2I models and other techniques (see Fig. 2), and a new Magnet pipeline figure for ease of understanding (see Fig. 3).

We have read all the papers mentioned by the reviewers and would like to highlight some differences as follows:

1. Existing works analyzing both word and EOT/pad embeddings emphasize the semantic effect of the word embedding, while we point out the entanglement of the pad embeddings to explain existing problems of SD (e.g., incorrect attributes, indistinguishable objects, see Appendix Fig. 13);

2. Related works require additional inputs or fine-tune the denoising U-Net, whereas Magnet takes only the given text as input and can be applied directly to any prompt;

3. Operating outside the U-Net, Magnet provides a plug-and-play capability and can be readily integrated to existing T2I models (e.g., SD, SDXL [1], PixArt [2]) and controlling techniques (e.g, Attend-and-Excite, layout-guidance [3], ControlNet [4]);

4. Magnet shows the anti-prior ability, i.e., to generate high-quality images with unnatural concepts, while related works are incompetent (see Appendix Fig. 24, and Fig. 2(e) compared to GORS [5] in the attached PDF).

Last but not least, we sincerely hope that our work will motivate the community to explore generative diffusion models and discover other interesting phenomena.

[1] SDXL: Improving Latent Diffusion Models for High-Resolution Image Synthesis

[2] PixArt-$\alpha$: Fast Training of Diffusion Transformer for Photorealistic Text-to-Image Synthesis

[3] Training-Free Layout Control with Cross-Attention Guidance

[4] Adding Conditional Control to Text-to-Image Diffusion Models

[5] T2I-CompBench: A Comprehensive Benchmark for Open-world Compositional Text-to-image Generation

---

### Decision · Program_Chairs · 2024-09-25

**Decision:**

Accept (poster)

**Comment:**

This paper investigates the attribute binding limitation in text-to-image diffusion models and introduces the Magnet approach, which enhances attribute-object alignment by interpolating object features with their designated attributes (positively) and attributes designated with other objects (negatively) specified in the prompt.

Summary Of Reasons To Publish:

1) Novel approach with interesting and in-depth analysis of the attribute binding problem

2) Well-organized and well-motivated work

3) Strong evidence of utility of proposed approach with human evaluations

Summary Of Suggested Revisions:

All the major concerns raised by the reviewers are addressed in the rebuttal to some extent. I suggest authors to incorporate the feedback on the presentation and clarity issues in the final version.